# An electrostatic selection mechanism controls sequential kinase signaling downstream of the T cell receptor

Neel H Shah[1,2,3], Qi Wang[1,2,3†], Qingrong Yan[1,2,3‡], Deepti Karandur[1,2,3], Theresa A Kadlecek[4,5], Ian R Fallahee[1,2,3], William P Russ[6], Rama Ranganathan[6,7,8], Arthur Weiss[4,5], John Kuriyan[1,2,3,9*]

[1]Department of Molecular and Cell Biology, University of California, Berkeley, United States; [2]California Institute for Quantitative Biosciences, University of California, Berkeley, United States; [3]Howard Hughes Medical Institute, University of California, Berkeley, United States; [4]Rosalind Russell/Ephraim P Engleman Rheumatology Research Center, Department of Medicine, University of California, San Francisco, United States; [5]Howard Hughes Medical Institute, University of California, San Francisco, United States; [6]Green Center for Systems Biology, University of Texas Southwestern Medical Center, Dallas, United States; [7]Department of Biophysics, University of Texas Southwestern Medical Center, Dallas, United States; [8]Department of Pharmacology, University of Texas Southwestern Medical Center, Dallas, United States; [9]Molecular Biophysics and Integrated Bioimaging Division, Lawrence Berkeley National Laboratory, Berkeley, United States

*For correspondence: kuriyan@berkeley.edu

Present address: †D E Shaw Research, New York, New York, United States; ‡Janssen Pharmaceutical Companies of Johnson and Johnson, Malvern, United States

**Abstract** The sequence of events that initiates T cell signaling is dictated by the specificities and order of activation of the tyrosine kinases that signal downstream of the T cell receptor. Using a platform that combines exhaustive point-mutagenesis of peptide substrates, bacterial surface-display, cell sorting, and deep sequencing, we have defined the specificities of the first two kinases in this pathway, Lck and ZAP-70, for the T cell receptor ζ chain and the scaffold proteins LAT and SLP-76. We find that ZAP-70 selects its substrates by utilizing an electrostatic mechanism that excludes substrates with positively-charged residues and favors LAT and SLP-76 phosphosites that are surrounded by negatively-charged residues. This mechanism prevents ZAP-70 from phosphorylating its own activation loop, thereby enforcing its strict dependence on Lck for activation. The sequence features in ZAP-70, LAT, and SLP-76 that underlie electrostatic selectivity likely contribute to the specific response of T cells to foreign antigens.

## Introduction

Signal transduction by the T cell receptor (TCR) triggers the activation of three non-receptor tyrosine kinases: Lck, ZAP-70, and Itk (*Smith-Garvin et al., 2009*; *Weiss and Littman, 1994*). A notable feature of this pathway is a strict hierarchy in kinase activation, which is accompanied by highly specific phosphorylation of substrates by each kinase (*Figure 1*). T cells must mount a strong and sustained response upon encountering a foreign peptide antigen bound to a major histocompatibility complex (MHC) molecule on an antigen-presenting cell, without launching an immune reaction against self-antigens. The origin of this selectivity is not well understood, and it cannot be explained by the modest differences in affinities between self and foreign peptide antigens for the T cell receptor

**eLife digest** A class of enzymes known as tyrosine kinases relay signals in cells by adding phosphate groups onto specific sites (called 'tyrosine residues') in other proteins. Most tyrosine kinases can phosphorylate many targets (or 'substrates'); they can also phosphorylate and thereby activate themselves, when given the right signal. Many tyrosine kinases select their substrates on the basis of their location; once recruited to and activated at a specific site, these enzymes will typically phosphorylate many nearby proteins.

A tyrosine kinase called ZAP-70 is found in immune cells known as T cells. ZAP-70 works together with another kinase called Lck to activate T cells, which enables the cells to mount an immune response when they encounter foreign molecules. This pathway is precisely controlled, with Lck activated first, followed by ZAP-70. Unlike most other tyrosine kinases, ZAP-70 cannot activate itself, and it will only phosphorylate a narrow range of substrates. The origin of these constraints are not understood, but they are thought to be crucial for ensuring that T cells readily respond to foreign molecules but not to healthy cells.

Shah et al. developed a high-throughput technique to investigate which features ZAP-70 and Lck use to select their substrates. First, hundreds of different sequences based on natural substrates were genetically encoded and introduced into bacterial cells, with one type per bacterium. The bacteria displayed these sequence variants on their surface, and Shah et al. then treated the bacteria with either ZAP-70 or Lck. Cell sorting was used to isolate those bacterial cells with variants that were phosphorylated, and high-throughput DNA sequencing was used to identify the phosphorylated sequences. This approach revealed that ZAP-70 was deterred from phosphorylating sites that carry a positive charge and strongly preferred sites that are negatively-charged, such as those found in its two major substrates. Shah et al. also showed that Lck, which behaves like a typical tyrosine kinase, could not phosphorylate the substrates of ZAP-70 because of their substantial negative charge. This lack of cross-reactivity between Lck and the ZAP-70 substrates prevents premature signaling in T cells.

Using simulations, Shah et al. went on to show that a positively-charged region on ZAP-70 (which is more prominent than in other tyrosine kinases) helps ZAP-70 interact with negatively-charged substrates. This region also deters the kinase from activating itself, making it dependent instead upon Lck for activation. Together, these results identify the distinctive features of ZAP-70 that are important for ensuring that T cells are activated only when they sense foreign molecules on unhealthy cells. The work will lead to future studies exploring the tightly controlled signaling events carried out by tyrosine kinases in T cells in more detail.

(*Palmer and Naeher, 2009*). The pattern of tyrosine kinase activity downstream of the T cell receptor is implicated in a kinetic proofreading mechanism that has been posited to underlie the fidelity of the T cell response (*Chakraborty and Weiss, 2014*; *McKeithan, 1995*). In such a mechanism, complexes formed between the T cell receptor and antigens must be sufficiently long-lived to propagate a biochemical signal through the sequential steps of kinase activation and substrate phosphorylation within the T cell.

The Src-family kinase Lck initiates intracellular signaling in T cells by phosphorylating the cytoplasmic tails of CD3 γ, δ, and ε, and the ζ subunits of the T cell receptor (*Figure 1*). This phosphorylation occurs on immunoreceptor tyrosine-based activation motifs (ITAMs), which are conserved sets of sequences bearing two tyrosines separated by ten to twelve residues (*Love and Hayes, 2010*; *Reth, 1989*). ZAP-70, a Syk-family kinase with tandem SH2 domains, is recruited to the membrane by binding these doubly-phosphorylated ITAMs (*Chan et al., 1991*; *Iwashima et al., 1994*; *Wange et al., 1993*). ZAP-70 is auto-inhibited before ITAM binding, and its activation requires that it be phosphorylated by Lck, both on its SH2-kinase linker and on the activation loop, a conserved regulatory segment in the kinase domain (*Deindl et al., 2007*; *Yan et al., 2013*).

Once activated, ZAP-70 phosphorylates two scaffold proteins, LAT and SLP-76 (*Bubeck Wardenburg et al., 1996*; *Au-Yeung et al., 2009*; *Zhang et al., 1998*). These scaffolds are phosphorylated at multiple sites, leading to the formation of phosphotyrosine-dependent signaling

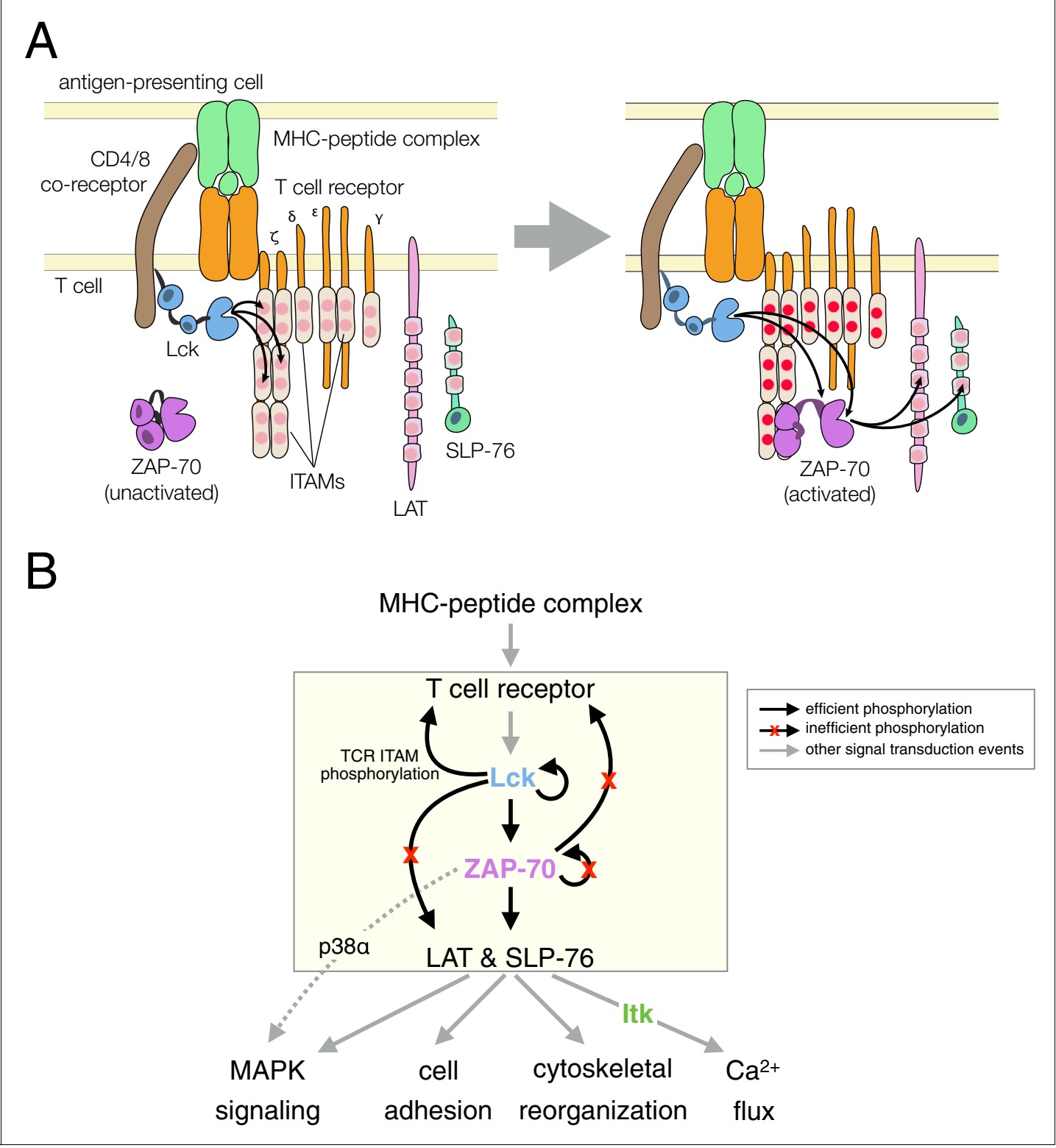

**Figure 1.** T cell receptor-proximal signaling. (**A**) Initiating events in T cell receptor signaling (based on *Au-Yeung et al., 2009*). (**B**) Ordered kinase activation and substrate phosphorylation by Lck and ZAP-70 leads to downstream signaling. Abbreviations: TCR, T cell receptor; MHC, major histocompatibility complex; Lck, lymphocyte-specific kinase; ZAP-70, ζ-chain associated protein of 70 kilodaltons; Itk, interleukin-2-inducible T cell kinase; ITAMs, immunoreceptor tyrosine-based activation motifs; LAT, linker for the activation of T cells; SLP-76, SH2-containing leukocyte protein of 76 kilodaltons; and MAPK, mitogen-activated protein kinase.

complexes (*Balagopalan et al., 2010*; *Zhu et al., 2003*). In one such complex, the Tec-family kinase Itk binds to phosphorylated SLP-76 through its SH2 domain (*Bunnell et al., 2000*). Itk phosphorylates and activates phospholipase-C$\gamma$1 (PLC$\gamma$1), which is bound to phosphorylated LAT. Signaling by PLC$\gamma$1 leads to a cytoplasmic calcium increase, thereby initiating part of the transcriptional response of the T cell to activation by antigens (*Weiss et al., 1991*).

The order of these signaling events is now well established (*Figure 1B*), and is thought to provide tight control over the T cell response. It is not clear, however, how this sequence of kinase reactions is enforced, given that tyrosine kinases are generally considered to be promiscuous and even 'sloppy' enzymes (*Mayer, 2012*). In this study, we focus on the key molecular features of ZAP-70, Lck, and their substrates that dictate the order of kinase activation and substrate phosphorylation in a T cell. Specifically, we examine why the substrate specificity of ZAP-70 is so narrow that it can only phosphorylate the scaffold proteins LAT and SLP-76 efficiently, and why ZAP-70 cannot phosphorylate and activate itself. We also ask why LAT and SLP-76 are not phosphorylated efficiently by Lck. The avoidance of phosphorylation of these two scaffold proteins by Lck is critical for ensuring that the more tightly regulated kinase, ZAP-70, propagates the signal downstream only after it is activated by Lck.

To answer these questions, we utilized a high-throughput mutagenesis and screening platform to measure kinase activity towards a large number of substrates simultaneously. Our approach generates sequence-activity relationships for variants of specific peptide substrates by genetically encoding scanning point-mutagenesis libraries of peptides into a bacterial surface-display system that has been developed previously (*Rice and Daugherty, 2008*). Phosphorylation of individual peptides in these libraries is detected by fluorescence-activated cell sorting (FACS), coupled to Illumina DNA sequencing technology (deep sequencing) (*Bentley et al., 2008*).

We applied this high-throughput platform to measure the effect of proximal mutations on the phosphorylation of several tyrosine residues in LAT and the T cell receptor $\zeta$ subunit (TCR$\zeta$). Our data show that the presence of multiple negatively-charged residues in the vicinity of tyrosines in LAT, and the exclusion of positively-charged ones, allows those tyrosines to be selected through electrostatic interactions with ZAP-70. This sequence pattern also prevents phosphorylation of LAT by Lck. Brownian dynamics and molecular dynamics simulations suggest that ZAP-70 detects the negative charge on LAT and SLP-76 by using a substrate binding region on its catalytic domain that is enriched in positively-charged residues to an extent that appears to be unique among non-receptor tyrosine kinases. Our analysis also suggests that the inability of ZAP-70 to undergo robust autophosphorylation is due to steric and electrostatic repulsion that blocks the adoption of the enzyme-substrate complex in which the activation loop of one kinase domain acts as the substrate for another. This feature appears to be a natural consequence of the specialization of ZAP-70 towards LAT and SLP-76 as preferred substrates.

## Results and discussion

### A high-throughput assay for tyrosine kinase specificity based on bacterial surface-display and deep sequencing

A widely-used approach to determine consensus sequences in kinase substrates relies on peptide libraries with random sequences surrounding a central tyrosine residue. In older versions of this technique, degenerate peptide libraries were treated with a kinase of interest, and the resulting phosphopeptides were separated and sequenced in bulk to determine the distribution of amino acid residues at each position (*Songyang et al., 1995*). Recent iterations of this technique utilize positional scanning peptide libraries where degenerate sub-libraries are created in which single amino acid residues are fixed at various positions flanking the tyrosine. All of the sub-libraries are phosphorylated in parallel and then immobilized on a membrane or microarray for high-throughput detection (*Deng et al., 2014*; *Hutti et al., 2004*). While these methods provide a reliable way to determine consensus motifs and can provide insights into kinase specificity (*Begley et al., 2015*), reliance on the analysis of degenerate mixtures prevents a direct comparison of the effects of specific mutations in particular substrate sequences.

To directly determine the importance of specific residues in substrates of ZAP-70 and Lck, we developed a high-throughput platform to measure the relative rates of phosphorylation for hundreds

of individual peptides simultaneously (*Figure 2*). This assay utilizes a previously developed scaffold for bacterial surface-display of peptides, the enhanced circularly permuted OmpX (eCPX) protein (*Rice and Daugherty, 2008*). This scaffold has been used to measure phosphorylation levels of tyrosine-containing peptides on the surface of *E. coli* cells upon addition of a tyrosine kinase to the cell suspension, followed by detection using a pan-phosphotyrosine antibody and flow cytometry (*Henriques et al., 2013*). We expanded this technique by applying it to libraries of genetically encoded peptides and coupling it to fluorescence-activated cell sorting (FACS), followed by deep sequencing (*Figure 2*).

In a typical experiment, *E. coli* cells were transformed with a DNA library encoding peptides fused to the eCPX scaffold. After growth and induction of scaffold expression, the cells were washed, then

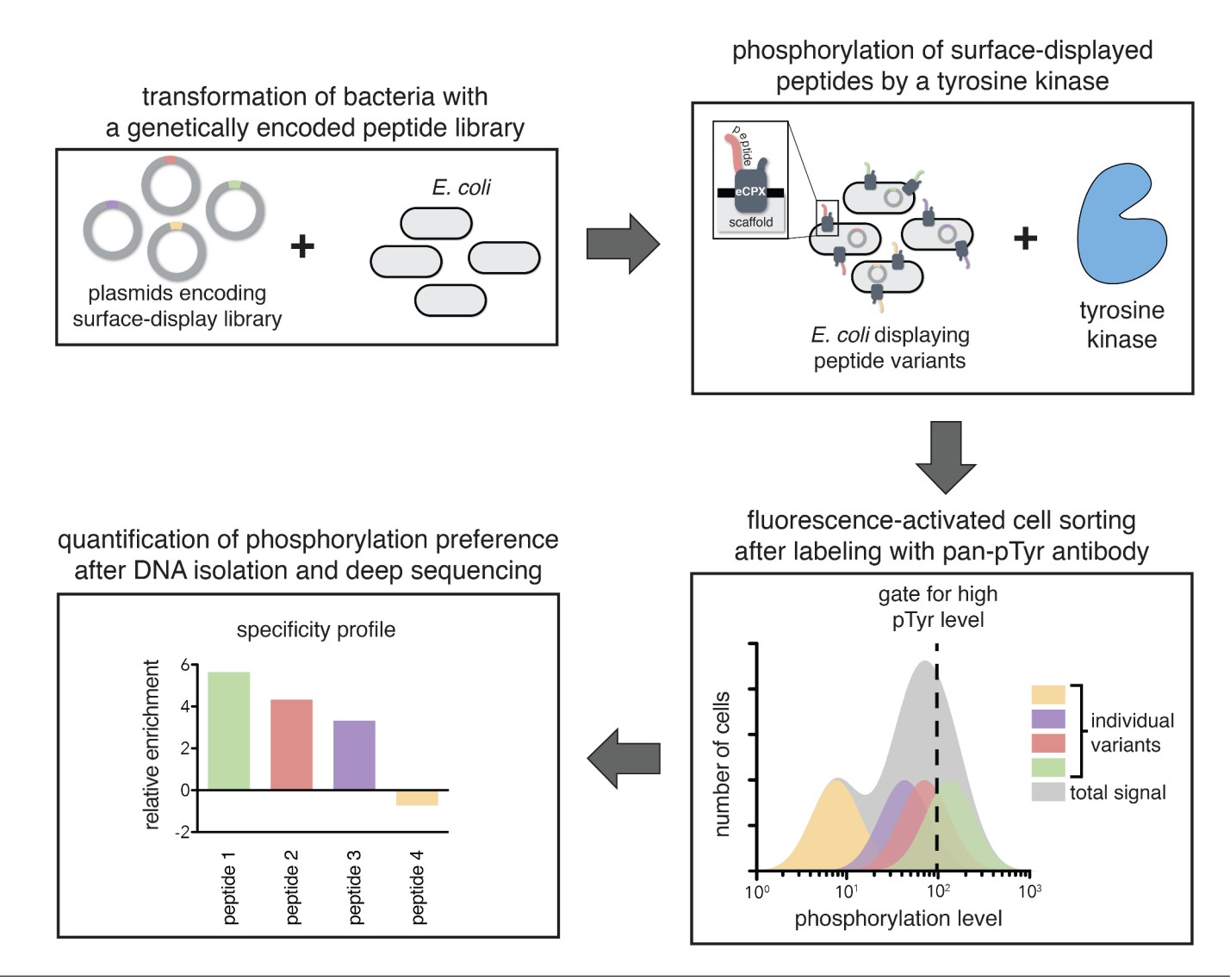

**Figure 2.** A high-throughput assay for tyrosine kinase specificity. Top left panel: *E. coli* cells are transformed with plasmids encoding a library of peptide variants fused to the bacterial surface-display scaffold, eCPX (*Rice and Daugherty, 2008*). Top right panel: Expression of the peptide-scaffold fusions is induced to permit surface-display of the peptides, then the peptides on the extracellular surface of the cells are phosphorylated by the addition of a tyrosine kinase to the cell suspension (*Henriques et al., 2013*). Bottom right panel: Phosphorylated cells are labeled with a fluorescent pan-phosphotyrosine antibody, and cells with a high fluorescence signal are isolated by fluorescence-activated cell sorting. Bottom left panel: DNA from the sorted cells and an unsorted control population is isolated and sequenced by Illumina deep sequencing to determine the enrichment of the DNA sequence encoding each variant in the library after selecting for a high phosphorylation level.

resuspended in a buffer with a tyrosine kinase, ATP, and $Mg^{2+}$. At an early time-point in the reaction, when it was less than 30% complete, the kinase activity was quenched by the addition of EDTA to the suspension. The cells were labeled with a fluorescent pan-phosphotyrosine antibody, and sorted for high phosphotyrosine level. DNA from both unsorted and sorted cells was isolated and deep-sequenced to determine the frequency of each peptide in the library before and after selection for high phosphorylation level.

For DNA corresponding to each peptide, an enrichment value was calculated as described previously for a high-throughput binding assay (*McLaughlin et al., 2012*). Briefly, the ratio of the abundance of DNA corresponding to a peptide in the sorted and unsorted samples was determined, and that enrichment ratio was normalized to the enrichment ratio for a reference member of the library. The normalized enrichment ratio for a particular DNA sequence in the library is a measure of the relative efficiency by which the corresponding peptide is phosphorylated by the kinase.

To test the validity of our approach, we first generated a small DNA library encoding the wild-type sequences of peptide segments from LAT, SLP-76, the putative ZAP-70 substrate p38α (*Salvador et al., 2005*), and TCRζ (see *Figure 3A* and *Figure 3—figure supplement 1* for sequences of the peptides used). The peptide segments were 19–22 residues long, and this library was screened against the isolated kinase domains of ZAP-70 and Lck (*Figure 3B*; see the appendix for a discussion of variation in peptide surface-display levels). The results of the initial screens recapitulated known specificity trends observed in T cells (*Chu et al., 1996*; *Isakov et al., 1996*; *Williams et al., 1998*). They were also consistent with in vitro measurements of kinetic parameters obtained using purified kinases and peptide substrates (*Figure 3—figure supplement 2*, *3*). Specifically, we found that ZAP-70 efficiently phosphorylates LAT and SLP-76, but not TCRζ ITAMs, while Lck is an efficient kinase for ITAMs, but not for LAT or SLP-76. Aside from Lck phosphorylation of LAT Tyr 64, neither kinase readily phosphorylated the first five tyrosine residues on LAT (*Figure 3B*), which are not known to be phosphorylated in T cells (*Balagopalan et al., 2010*). The relative phosphorylation rates for three of the phosphosites in our initial screen (Tyr 132 and Tyr 191 of LAT and Tyr 145 of SLP-76) have been measured in T cells, and those results are consistent with our peptide-based measurements (*Houtman et al., 2005*).

## An electrostatic mechanism for the selection of substrates by ZAP-70

The obvious common feature of the known phosphorylation sites for ZAP-70 is that each tyrosine is surrounded by several acidic residues, with only the rare positively-charged residue nearby (*Figure 3A*). By contrast, Lck substrates typically have both positively- and negatively-charged residues near the tyrosine, and they have a near-neutral net charge. This suggests that high negative charge is likely to be a key determinant of whether a potential target site is phosphorylated by ZAP-70, but the location and number of negatively-charged residues around each of the tyrosines in LAT and SLP-76 is not conserved. There may also be other roles for the negative charge on LAT and SLP-76 that are unrelated to kinase specificity. For example, it was shown recently that LAT phosphorylation results in clustering, and that the phosphatase CD45 is excluded from these LAT clusters because it is negatively-charged (*Bunnell et al., 2002*; *Su et al., 2016*).

To define the sequence determinants of efficient phosphorylation by ZAP-70, we created three scanning point-mutagenesis libraries based on LAT sequences spanning Tyr 127, Tyr 132, and Tyr 226. In these libraries, every possible single mutation in a 20 residue peptide segment was represented with near-equal stoichiometry, as verified by DNA sequencing. Each library was screened against the ZAP-70 kinase domain in triplicate or quadruplicate (*Figures 4* and *5*). These data allow us to assess the impact of all possible point mutations at a single site on a given peptide, as shown for substitutions of Asp 225 and Glu 231 in the LAT Tyr 226 peptide (*Figures 4A and B*). The data also reveal the impact of individually introducing any particular amino acid residue at all sites within that peptide, as shown for lysine substitutions at each position in the LAT Tyr 226 peptide (*Figure 4C*). The full datasets from each screen are represented as heatmaps, which display the impact of all possible individual substitutions at every site in the peptide (*Figure 5*).

The results of these screens were highly reproducible (*Figure 5—figure supplement 1*), the trends were independent of the specific pan-phosphotyrosine antibody used, and enrichment values showed a strong correlation with measurements of phosphorylation rates for purified peptides (*Figure 5B*). In all three screens, mutation of the target tyrosine residues (Tyr 226, Tyr 127, and Tyr 132, respectively) resulted in a substantial depletion of DNA encoding those peptides after sorting

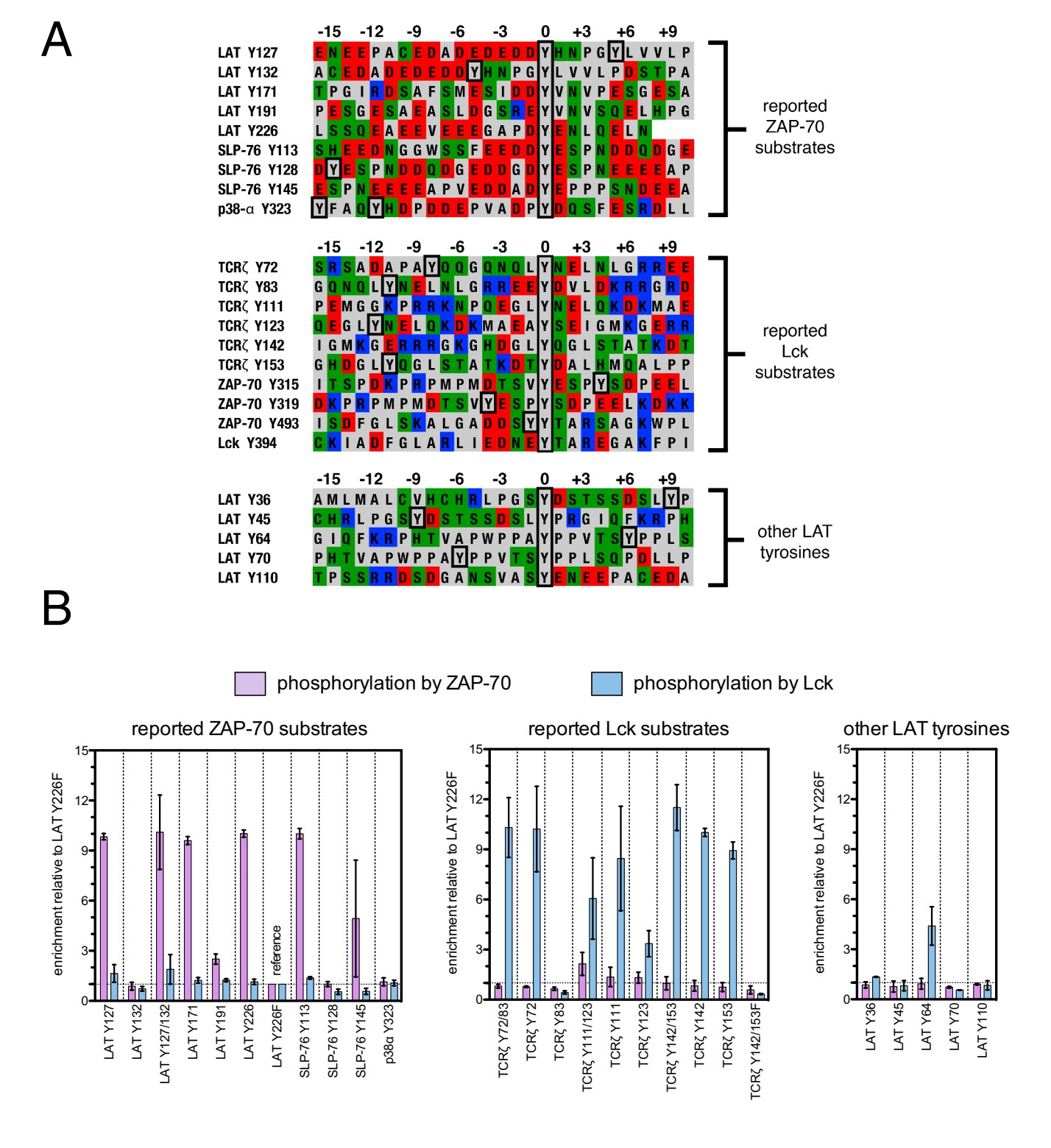

**Figure 3.** Phosphorylation by ZAP-70 or Lck of a variety of peptides based on LAT, SLP-76, p38α, and TCRζ. (A) Sequences surrounding the key tyrosine residues analyzed in this study. The focal tyrosine in each sequence is denoted as residue 0, and other proximal tyrosines are highlighted by a black box. (B) Enrichment of peptides from a library of peptide sequences based on LAT, SLP-76, p38α, and TCRζ after phosphorylation by ZAP-70 or Lck, followed by selection. Error bars represent standard deviations in enrichment values from two screens. The exact peptide sequences for each

*Figure 3 continued on next page*

*Figure 3 continued*

member of the library analyzed in panel B, including the locations of several tyrosine-to-phenylalanine mutants, are given in *Figure 3—figure supplement 1*.

The following figure supplements are available for figure 3:

**Figure supplement 1.** List of peptide sequences in the library containing segments of LAT, SLP-76, p38α, and TCRζ.

**Figure supplement 2.** List of purified peptides used for in vitro kinetic assays.

**Figure supplement 3.** In vitro phosphorylation kinetics of purified peptides by the ZAP-70 and Lck kinase domains.

(negative enrichment value relative to wild-type), setting a 'floor' for the detection limit of the assay (*Figure 5A,C and D*). The relative magnitudes of the kinetic effects and the bacterial display selectivity effects (the slope of the line in *Figure 5B*) depend on the peptide being tested. The data shown in *Figure 5B* are for the peptide spanning LAT Tyr 226, a good substrate for ZAP-70, and individual substitutions only have a modest, albeit measurable, effect on phosphorylation rate. By contrast, for ZAP-70 phosphorylation of a poor substrate, such as the peptide encompassing LAT Tyr 132 (*Figure 5D*), substitutions at Gly 131 result in as much as a 16-fold enhancement in phosphorylation rate (*Figure 5—figure supplement 4*).

A striking result from these screens is that the selection of substrates by ZAP-70 is controlled by an electrostatic filter, whereby the presence of a lysine or arginine residue anywhere within seven residues upstream or downstream of the substrate tyrosine severely compromises the efficiency of phosphorylation. A similar, but less dramatic, reduction in phosphorylation efficiency was observed when tyrosine-proximal residues were substituted by histidine. Replacement of native negatively-charged residues by neutral residues had a mild detrimental effect on phosphorylation efficiency, and introduction of acidic residues at neutral positions was often beneficial (*Figure 5*). Consistent with these observations, replacement of a single glutamate residue by lysine, either upstream or downstream of the tyrosine, resulted in a 30–40% reduction in the LAT Tyr 226 phosphorylation rate by ZAP-70 in kinase assays using purified peptide substrates, and mutation of multiple glutamates to alanine or lysine residues further reduced phosphorylation (*Figure 5—figure supplement 2*). The deleterious effect of arginine and lysine residues at any position also explains the slow phosphorylation of LAT Tyr 191 by ZAP-70 (*Figure 3—figure supplement 3*). Replacement of Arg 189 (at the −2 position) by alanine or aspartate residues successively improved the efficiency of Tyr 191 phosphorylation by ZAP-70 (*Figure 5—figure supplement 3*).

In addition to revealing the exclusion rule for arginine and lysine, the screens also emphasize the importance of particular amino acid residues at three specific positions. ZAP-70 has a very strong preference for an aspartate residue at the −1 position relative to the tyrosine residue and a modest preference for a glutamate or hydrophobic residue at the +1 position, consistent with known preferences for the ZAP-70 paralog Syk (*Deng et al., 2014*; *Schmitz et al., 1996*; *Xue et al., 2012*). There is also a preference for a hydrophobic residue at the +3 position in all three peptides, which is a feature common to other tyrosine kinases (*Bose et al., 2006*; *Deng et al., 2014*).

Given the strong preference for an aspartate residue at the −1 position, we were intrigued that the residue immediately before Tyr 132 in LAT is a glycine. Phosphorylation of Tyr 132 is essential for T cell differentiation and function, as it generates the docking site for PLCγ1, which initiates calcium signaling (*Roncagalli et al., 2010*). In our screen for Tyr 132 phosphorylation (*Figure 5D*), replacement of Gly 131 with virtually any other residue improved the efficiency of phosphorylation by ZAP-70, and the G131D mutation caused a 16-fold enhancement of phosphorylation by ZAP-70 in kinase assays using purified peptides (*Figure 5—figure supplement 4*). Gly 131 in LAT is highly conserved in mammals, and it is also common in other vertebrates (*Figure 5—figure supplement 5*). PLCγ1 binding sites on other proteins do not contain a glycine adjacent to the phosphotyrosine, indicating that this is not a requirement for SH2 domain engagement (*Jones et al., 2006*; *Leung et al., 2014*). A slow rate of phosphorylation at Tyr 132 in LAT may be important for kinetic proofreading during T cell receptor signaling, the significance of which will be explored in future studies.

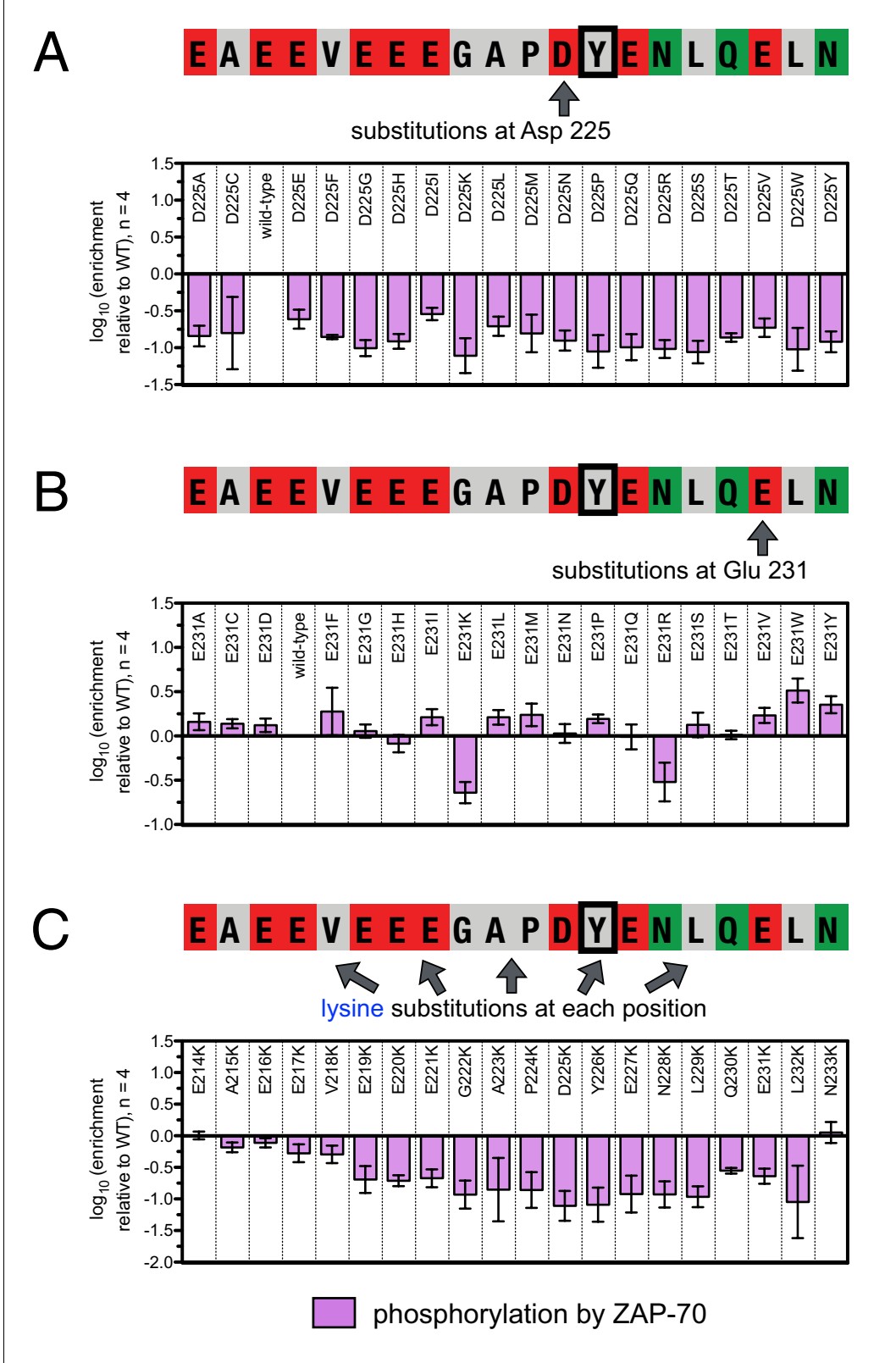

**Figure 4.** Effect of single amino acid substitutions on the phosphorylation of LAT Tyr 226 by ZAP-70. (**A**) Impact of all amino acid substitutions at LAT Asp 225 on the phosphorylation of Tyr 226 by ZAP-70. (**B**) Impact of all amino acid substitutions at LAT Glu 231 on the phosphorylation of Tyr 226 by ZAP-70. (**C**) Impact of lysine substitutions at all 20 positions (LAT residues 214–233) on the phosphorylation of Tyr 226 by ZAP-70. Error bars represent standard deviations from an average of four measurements. The enrichment values of each variant were normalized to the enrichment value of the wild-
*Figure 4 continued on next page*

Figure 4 continued

type sequence and presented on a logarithmic scale. Thus, a value of 0 indicates that the substitution did not impact phosphorylation relative to the wild-type sequence, whereas positive and negative values mean that the substitution enhanced or diminished phosphorylation, respectively.

## The substrate sequence preferences of Lck are distinct from those of ZAP-70

To understand why Lck readily phosphorylates ITAM peptides but not LAT- or SLP-76-based peptides, we screened three libraries against Lck in our high-throughput platform (*Figure 6*). Two libraries were based on the second ITAM in TCRζ, one spanning Tyr 111 in a Y123F background, and the other spanning Tyr 123 in a Y111F background (*Figures 6A and B*, respectively). The two ITAM libraries have the tyrosine close to one or the other terminus of the peptide. Thus, each library provides information on Lck specificity either downstream or upstream of the tyrosine residue, respectively. We also screened one of the LAT point mutant libraries (LAT Tyr 127) against the Lck kinase domain (*Figure 6C*). We chose this library because the sequence surrounding LAT Tyr 127 has the canonical features of most ZAP-70 substrates (many acidic residues, no lysines or arginines, and a −1 aspartate), but in vitro measurements indicated that it was still a modest substrate for Lck (*Figure 3—figure supplement 3*).

Analysis of these three libraries revealed three major differences between ZAP-70 and Lck: (1) In contrast to the lysine/arginine exclusion rule for ZAP-70, Lck has a modest preference for positively-charged residues, (2) Lck does not have a requirement for an aspartate residue at the −1 position, and (3) ZAP-70 recognizes residues both upstream and downstream of the phosphosite, while Lck largely utilizes downstream residues for substrate discrimination. This point is important for understanding the difference in autophosphorylation ability between the two kinases, as discussed below.

Lck has a strong preference for a bulky hydrophobic residue at the −1 position relative to the phosphosite, and it does not tolerate an aspartate residue at this position. This preference explains why Lck does not readily phosphorylate tyrosine residues in LAT and SLP-76 that contain an aspartate residue at the −1 position (*Figure 3*). It also explains why Lck phosphorylates the first tyrosine residue in each ITAM faster than the second one (*Housden et al., 2003*), as the first tyrosine is typically preceded by a leucine while the second is preceded by a smaller aliphatic or polar residue (*Figure 3A*).

The lysine/arginine exclusion rule observed for ZAP-70 substrate sequences clearly does not apply to Lck. Mutation of acidic residues located upstream of Tyr 127 in LAT to virtually any other residue, including lysine and arginine, resulted in a slight, but significant, enhancement of phosphorylation efficiency by Lck (*Figure 6C*). Downstream of the tyrosine, lysine and arginine residues were not tolerated at the +1 to +3 positions, but positively-charged residues were preferred at positions located further downstream (*Figure 6D*). This is consistent with the presence of several lysine and arginine residues located five to ten positions downstream of the phosphosites in TCRζ ITAMs (*Figure 3A*), and mutation of native lysine residues downstream of TCRζ Tyr 111 to aspartate or glutamate was detrimental for Tyr 111 phosphorylation by Lck (*Figure 6A*).

To validate our conclusions, we measured the phosphorylation kinetics of purified mutant LAT peptides by Lck. Simultaneous mutation to alanine of several glutamates located five to seven residues upstream of LAT Tyr 226 had no measurable effect on phosphorylation by Lck, and mutation of those residues to lysine only reduced the rate of phosphorylation modestly (*Figure 6—figure supplement 1*). The same mutations to lysine in LAT reduced the rate of Tyr 226 phosphorylation by ZAP-70 by more than 80% (*Figure 5—figure supplement 2*). Mutation of Glu 231 (+5 position) to alanine caused a two-fold increase in the Lck-catalyzed rate of LAT Tyr 226 phosphorylation, suggesting that a downstream negative charge on substrates might repel Lck, and this rate was further enhanced by mutation of that same glutamate residue to lysine (*Figure 6—figure supplement 1*). By contrast, the E231K mutation had a negative impact on LAT Tyr 226 phosphorylation by ZAP-70 (*Figure 5—figure supplement 2*). These observations explain why Lck phosphorylates LAT Tyr 127 with moderate efficiency: this is the only ZAP-70 substrate without a negatively-charged residue downstream of the tyrosine (*Figure 3A*).

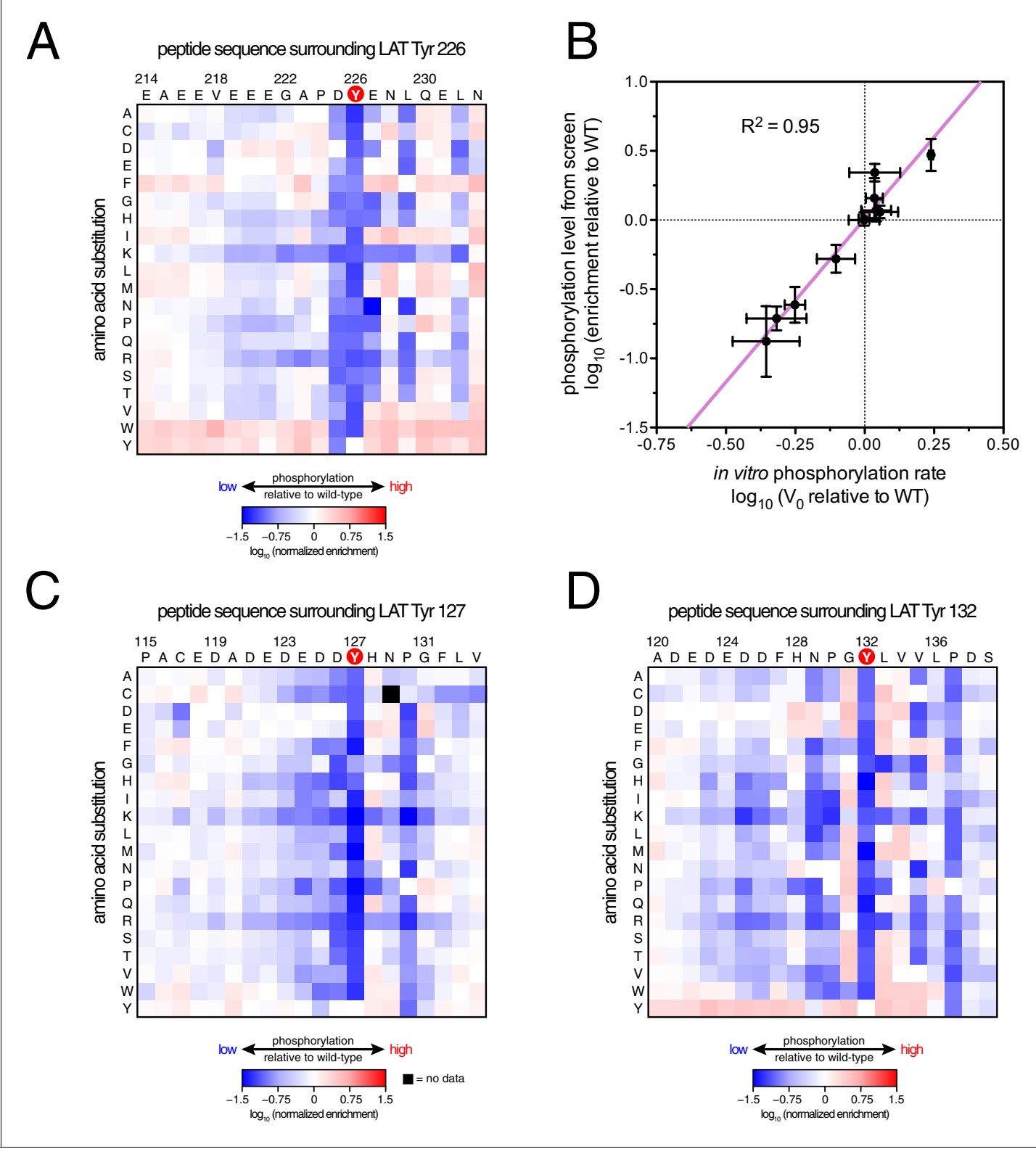

**Figure 5.** ZAP-70 phosphorylation of LAT point mutant libraries. (**A**) Average enrichment values from four independent screens for phosphorylation of the LAT Tyr 226 library by ZAP-70. (**B**) Correlation between the rates of phosphorylation of 11 purified LAT Tyr 226 variants and their enrichment values from the screen shown in panel A. Horizontal error bars represent the standard deviations from three kinetic measurements with purified peptides, and vertical error bars represent the standard deviations from four screens. (**C**) Average enrichment values from three independent screens for

*Figure 5 continued on next page*

*Figure 5 continued*

phosphorylation by ZAP-70 of the LAT Tyr 127 library in a Y132F background. (D) Average enrichment values from three independent screens for phosphorylation by ZAP-70 of the LAT Tyr 132 library in a Y127F background. All enrichment values are $\log_{10}$-transformed and normalized relative to the parent peptide sequence in that screen, which has a value of 0.

The following figure supplements are available for figure 5:

**Figure supplement 1.** Correlations between enrichments from four replicates for ZAP-70 phosphorylation of the point mutant library spanning LAT Tyr 226.

**Figure supplement 2.** Phosphorylation by the ZAP-70 kinase domain of peptides spanning LAT Tyr 226 with various charge-altering substitutions.

**Figure supplement 3.** Phosphorylation by the ZAP-70 kinase domain of peptides spanning LAT Tyr 191 with the R189A or R189D substitutions.

**Figure supplement 4.** Phosphorylation by the ZAP-70 kinase domain of peptides spanning LAT Tyr 132 with or without the G131D substitution.

**Figure supplement 5.** Sequence logos based on an alignment of vertebrate LAT segments surrounding Tyr 132.

## ZAP-70 substrate recognition is governed by a positively-charged region surrounding the kinase active site

Unlike Lck, ZAP-70 has a relatively large positively-charged patch surrounding the binding site for peptide substrates, which provides potential contact points for substrate residues both upstream and downstream of the tyrosine (*Figures 7A and B*). This region in ZAP-70 has at least ten lysine and arginine residues and only three negatively-charged residues, one of which is required for catalysis (Asp 461). To identify interactions that might dictate substrate recognition, we ran molecular dynamics simulations of the ZAP-70 kinase domain bound to a peptide spanning LAT residues 219 to 233 (surrounding Tyr 226). Using the structure of the substrate-bound insulin receptor kinase domain as a guide (*Hubbard, 1997*; *Parang et al., 2001*), we built a model of the ZAP-70 kinase domain in an active conformation, bound to the LAT peptide, ATP, and two Mg$^{2+}$ ions (there are no crystal structures of substrate complexes of ZAP-70). The peptide was modeled into the active site by docking the tyrosine residue and the three residues immediately after it so that they form an antiparallel β sheet with the activation loop, as seen in tyrosine kinase-substrate co-crystal structures (*Bose et al., 2006*; *Hubbard, 1997*; *Levinson et al., 2006*; *Parang et al., 2001*; *Zhang et al., 2006*). The remaining peptide residues were modeled in an arbitrary conformation, projecting into the solvent and away from the kinase domain (*Figure 7—figure supplement 1*).

Using this starting structure, we generated five molecular dynamics trajectories that ran for 500 ns each (*Supplementary files 1* and *2*). In all five simulations, the short antiparallel β sheet formed by the peptide and the activation loop persisted for the length of each trajectory (*Figure 7—figure supplement 2*). Additionally, the phospho-acceptor tyrosine residue often adopted a configuration in close proximity to the catalytic aspartate residue (Asp 461) and the γ-phosphate of ATP, consistent with the requirements for phospho-transfer (*Figure 7—figure supplement 3*).

The molecular dynamics simulations revealed the formation of a stable ion pair between the −1 aspartate of the substrate (Asp 225) and Lys 504, a highly conserved residue in tyrosine kinases (*Figure 7C* and *Figure 7—figure supplement 4*). Located near the active site, and within a substrate-binding site that leads into it, is a loop connecting helices F and G of the kinase domain (the FG loop, *Figure 7B*). This loop presents four lysine residues towards the substrate (residues 538, 541, 542, and 544). The −1 Asp frequently formed an interaction with Lys 538 and made occasional contact with Lys 544 (*Figure 7C* and *Figure 7—figure supplement 4*). In Lck, the residues in the FG loop corresponding to Lys 538 and Lys 544 in ZAP-70 are isoleucine and threonine, respectively, which presumably removes the requirement for Asp at the −1 position of the substrate and enforces the preference for hydrophobic residues that has been noted previously (*Figure 6*) (*Songyang et al., 1995*).

The molecular dynamics simulations show that the FG loop in ZAP-70 is also important for recognition of residues located at the −5 to −7 positions with respect to the tyrosine residue (*Figure 7C*). For the LAT Tyr 226 peptide, the upstream residues Glu 219, Glu 220, and Glu 221 formed frequent

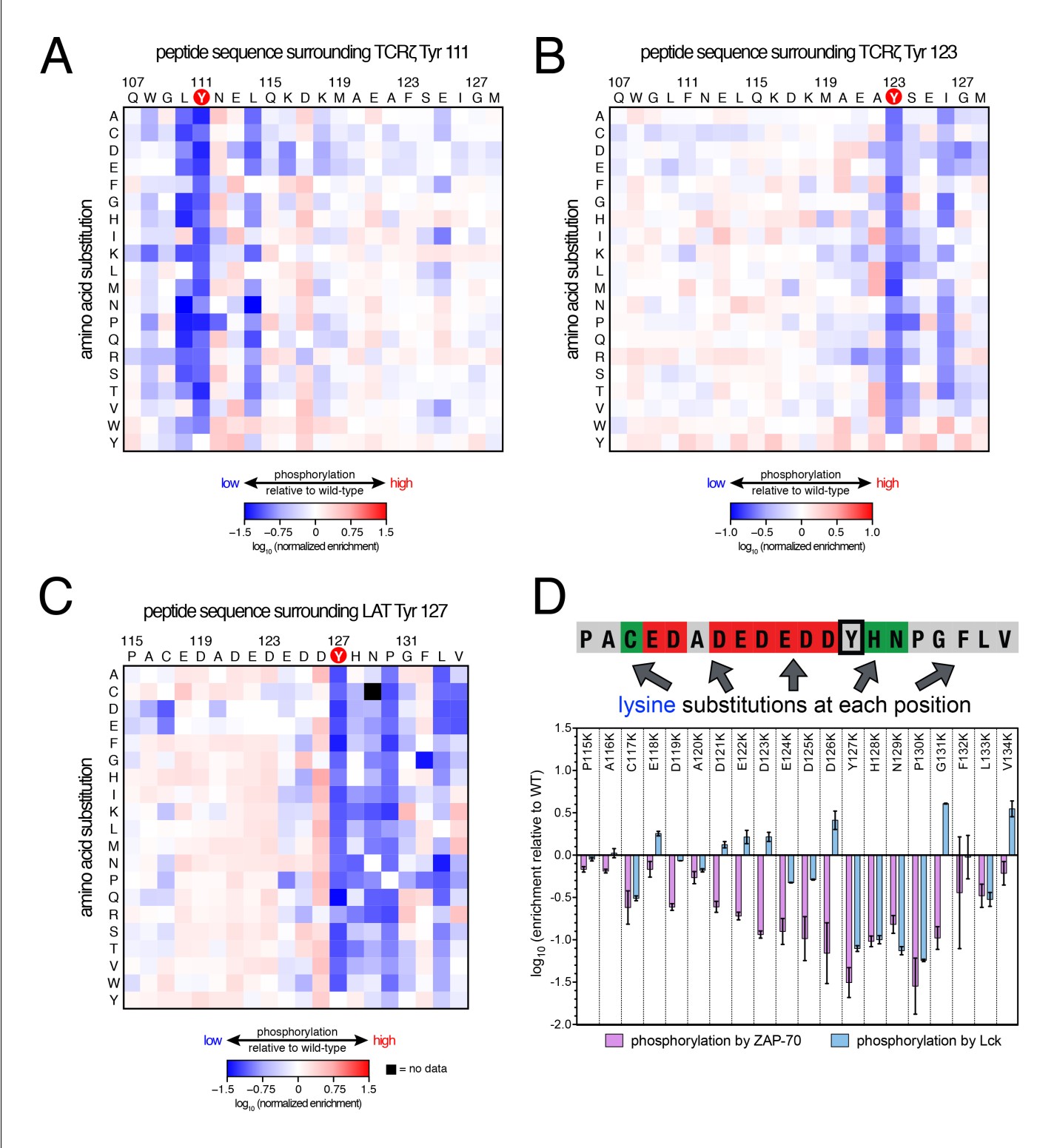

**Figure 6.** Lck phosphorylation of TCRζ ITAM and LAT point mutant libraries. (**A**) Average enrichment values from two independent screens for phosphorylation by Lck of the TCRζ Tyr 111 library in a Y123F background. (**B**) Average enrichment values from two independent screens for phosphorylation by Lck of the TCRζ Tyr 123 library in a Y111F background. (**C**) Average enrichment values from two independent screens for phosphorylation by Lck of the LAT Tyr 127 library in a Y132F background. (**D**) Comparison of the effect of lysine mutations on LAT Tyr 127 phosphorylation by ZAP-70 and Lck. Error bars represent standard deviations in enrichment values from three and two screens with ZAP-70 and Lck, respectively. All enrichment values are log₁₀-transformed and normalized relative to the parent peptide sequence in that screen, which has a value of 0.
*Figure 6 continued on next page*

*Figure 6 continued*

The following figure supplement is available for figure 6:

**Figure supplement 1.** Phosphorylation by the Lck kinase domain of peptides spanning LAT Tyr 226 with various charge-altering substitutions.

but transient ion pairs with all four lysine residues on the FG loop as well as with Lys 504. These ion pairs typically persisted for 5 to 10 ns before breaking and forming again. While any specific ion pair was short-lived, the clusters of negative charges on LAT and positive charges on ZAP-70 ensured that at least one or more ion pairs were formed in at least 50% of the instantaneous structures sampled from each trajectory (*Figure 7—figure supplement 5*).

The simulations also showed that ZAP-70 recognizes negatively-charged residues located downstream of Tyr 226. In the molecular dynamics trajectories, Glu 231, located five residues downstream of Tyr 226, frequently formed an interaction with Arg 514 in ZAP-70 (*Figure 7C* and *Figure 7—figure supplement 6*). In Lck, the residue corresponding to this arginine is a glycine, which explains why Lck does not select for negative residues at this position in the substrate.

The results of the molecular dynamics simulations are consistent with a number of experimental measurements of kinase activity. We purified a ZAP-70 kinase domain construct in which three of the four lysine residues in the FG loop were replaced by alanine (a 'KKMK-to-AAMA' mutation in the FG loop). This mutant ZAP-70 displayed substantially reduced activity against most of the LAT and SLP-76 peptides in kinase assays, and also showed an increased ability to phosphorylate ITAMs, when compared to wild-type ZAP-70 (*Figure 7—figure supplement 7*). When upstream glutamate residues in a LAT Tyr 226-containing peptide were replaced by alanine or lysine, there was a marked reduction in phosphorylation efficiency by wild-type ZAP-70. By contrast, the phosphorylation rate for ZAP-70 bearing the KKMK-to-AAMA mutation was not affected by these charge-altering mutations in LAT (*Figure 7—figure supplement 8*).

Arg 514 in ZAP-70 interacts with negatively-charged residues downstream of the tyrosine, and the importance of this residue for the ability of ZAP-70 to phosphorylate LAT was validated by mutation of this residue to alanine and glutamate. Both mutations reduced the rate at which ZAP-70 phosphorylated LAT Tyr 226. Replacing Glu 231 (+5 position) in the LAT peptide with lysine reduced phosphorylation by wild-type ZAP-70, but the presence of the lysine in the substrate peptide had no effect on phosphorylation by ZAP-70 with Arg 514 mutated to alanine or glutamate, consistent with the removal of a repulsive interaction (*Figure 7—figure supplement 9*). The E231K mutation in the LAT Tyr 226 peptide results in reduced phosphorylation by ZAP-70. In contrast, for Lck, this mutation results in increased phosphorylation (*Figure 5—figure supplement 2* and *Figure 6—figure supplement 1*). This is consistent with the opposite charge preferences of ZAP-70 and Lck downstream of substrate tyrosines (*Figure 6*).

## Brownian dynamics simulations suggest a role for long-range electrostatic effects in modulating the specificity of ZAP-70

While the molecular dynamics simulations point to interactions between ZAP-70 and LAT that are important once the peptide is bound, we wondered if long-range electrostatic interactions could also play a role in substrate recognition. We carried out Brownian dynamics simulations to address this question (*Gabdoulline and Wade, 1998*; *Northrup and Erickson, 1992*). In these simulations, the diffusive motions of molecules can be tracked over much longer timescales than in conventional molecular dynamics, because internal motions and the detailed structure of the solvent are neglected (*Figure 8A*). Brownian dynamics trajectories were calculated using the program SDA (*Martinez et al., 2015*), and the kinase domains and the peptide substrates were both treated as rigid bodies that interact only through electrostatic and van der Waals forces. The effects of water molecules and ions were modeled through the Poisson-Boltzmann equation for continuum electrostatics (*Baker et al., 2001*).

We generated Brownian dynamics trajectories using crystal structures of the kinase domains of ZAP-70 and Lck (PDB codes 1U59 (*Jin et al., 2004*) and 1QPJ (*Zhu et al., 1999*), respectively). Peptides were modeled as rigid and extended β strands of 15 residues, with 7 residues on either side of

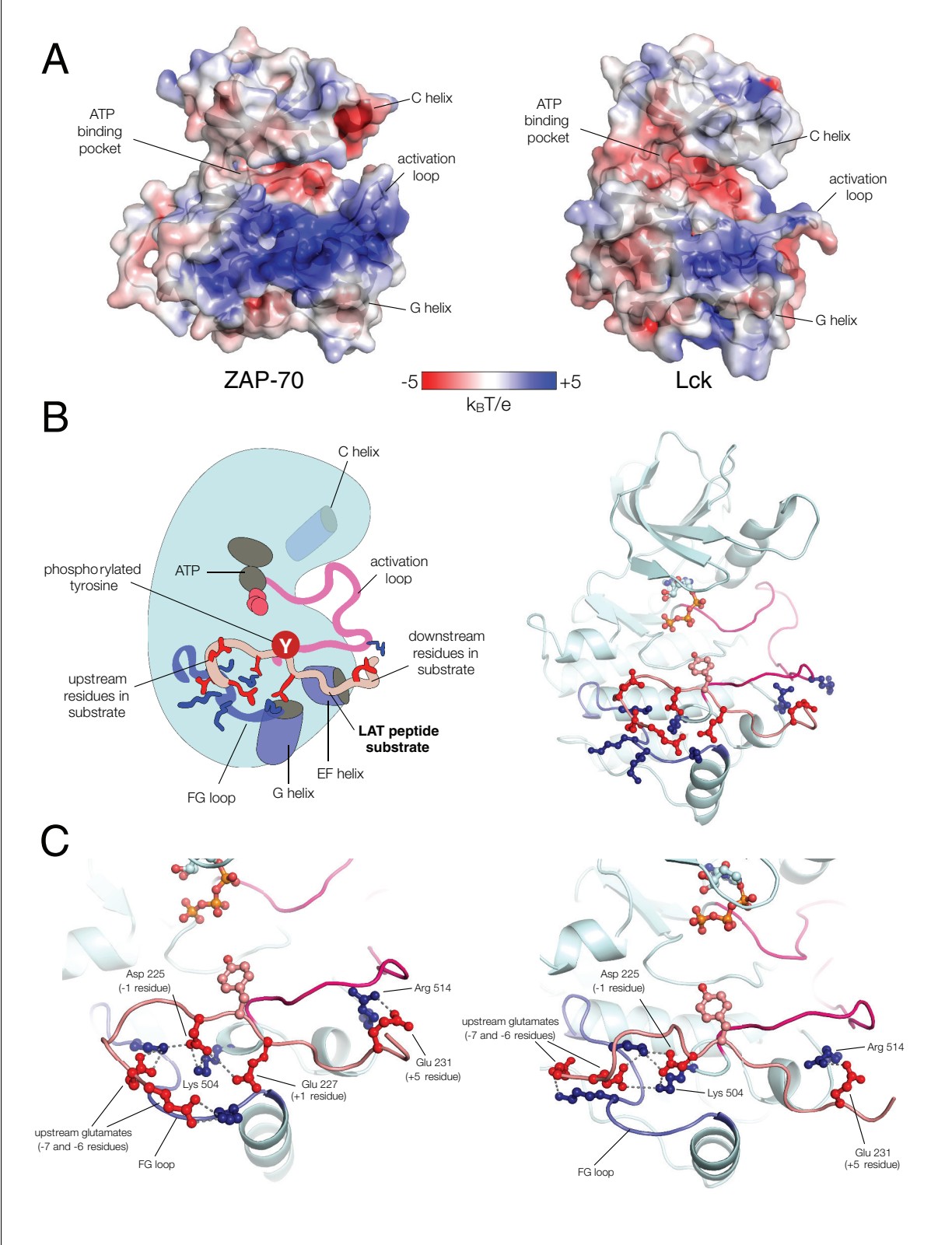

**Figure 7.** Electrostatic features of the ZAP-70 and Lck kinase domains, and peptide binding modes of ZAP-70. (A) The electrostatic surface potential, calculated using APBS (*Baker et al. 2001*) and displayed using PyMOL (*Schrödinger 2015*), of the ZAP-70 kinase domain (PDB code 1U59) and the Lck kinase domain (PDB code 1QPJ). (B) Schematic representation (left) and an instantaneous structure from a molecular dynamics simulation (right) of the

*Figure 7 continued on next page*

*Figure 7 continued*

ZAP-70 kinase domain bound to a peptide containing LAT Tyr 226. (C) Two different instantaneous structures from molecular dynamics simulations of ZAP-70 bound to the same peptide, highlighting key kinase-substrate interactions.

The following figure supplements are available for figure 7:

**Figure supplement 1.** Starting structure used for molecular dynamics simulations of ZAP-70 bound to a peptide spanning LAT Tyr 226.

**Figure supplement 2.** Hydrogen bonds between the ZAP-70 activation loop and the peptide +1 and +3 positions during five molecular dynamics trajectories.

**Figure supplement 3.** The close proximity between the LAT phospho-acceptor tyrosine and ATP (left) or the catalytic aspartate in ZAP-70 (right) during five molecular dynamics trajectories.

**Figure supplement 4.** Ion pairs between LAT Asp 225 (−1 position) and lysine residues 504 and 538 on ZAP-70 during five molecular dynamics trajectories.

**Figure supplement 5.** The frequency of ion pairs observed during instantaneous structures sampled every nanosecond from each trajectory.

**Figure supplement 6.** Ion pairs between LAT Glu 231 (+5 position) and Arg 514 on ZAP-70 during five molecular dynamics trajectories.

**Figure supplement 7.** In vitro phosphorylation kinetics of purified peptides by the wild-type ZAP-70 kinase domain and an FG loop mutant containing the K541A, K542A, and K544A substitutions.

**Figure supplement 8.** In vitro phosphorylation kinetics of purified peptides surrounding LAT Tyr 226 with various upstream charge-altering mutations by the wild-type ZAP-70 kinase domain and an FG loop mutant containing the K541A, K542A, and K544A substitutions.

**Figure supplement 9.** In vitro phosphorylation kinetics of purified peptides surrounding LAT Tyr 226 with the E231A or E231K mutations by the wild-type ZAP-70 kinase domain and mutants containing the R514A or R514E substitutions.

the phospho-acceptor tyrosine, with sequences corresponding to various LAT phosphosites. For each kinase-peptide pair, 10,000 trajectories were initiated by randomly placing the peptide on the surface of a sphere of radius 150 Å, centered on the center of mass of the kinase domain (*Figure 8A*). A Brownian dynamics trajectory was then generated for each initial configuration of the peptides. Trajectories were terminated when the peptide moved more than 250 Å from the center of mass of the protein. For analysis, coordinates were sampled from the trajectories every 200 ps, leading to ~1 million to ~100 million instantaneous structures for the different peptides (peptides that tend to stay longer in the vicinity of the protein lead to longer trajectories).

To visualize the Brownian dynamics trajectories for each peptide, we selected all instantaneous structures for which the Cα atom of the tyrosine residue is within 70 Å of the center of mass of the protein. A random sampling of 0.1% of these peptide positions is shown in *Figure 8B* for six different systems, five for ZAP-70 with peptide segments corresponding to Tyr 127, Tyr 226, Tyr 171, Tyr 110, and Tyr 191 in LAT, and one for Lck with the peptide corresponding to Tyr 127 in LAT. The first three tyrosine residues are good substrates for ZAP-70, while Tyr 110 and Tyr 191 are relatively poor ones (*Figure 3*). Tyr 127 and Tyr 226 both have a net charge of −6, with the charges on Tyr 171, Tyr 110, and Tyr 191 being −4, −3, and −2, respectively.

Comparison of the number of instantaneous structures within the 70 Å sphere for the different peptides is instructive – the greater this number, the longer the peptide stayed in the vicinity of the kinase domain before escaping beyond the 250 Å horizon. By this metric, Tyr 127, a good ZAP-70 substrate, is enriched around the ZAP-70 kinase domain by a factor of ~150 compared to Tyr 191, a poor substrate. Also, Tyr 127 is enriched around the ZAP-70 kinase domain by a factor of ~200 relative to the same peptide around Lck.

Within the 70 Å sphere, the good substrates for ZAP-70 are preferentially localized around the active site. This effect is particularly striking for Tyr 127 and Tyr 226 (*Figure 8B*, top-left and top-middle panels). Tyr 191, which is a poor substrate for ZAP-70, does not cluster around the ZAP-70

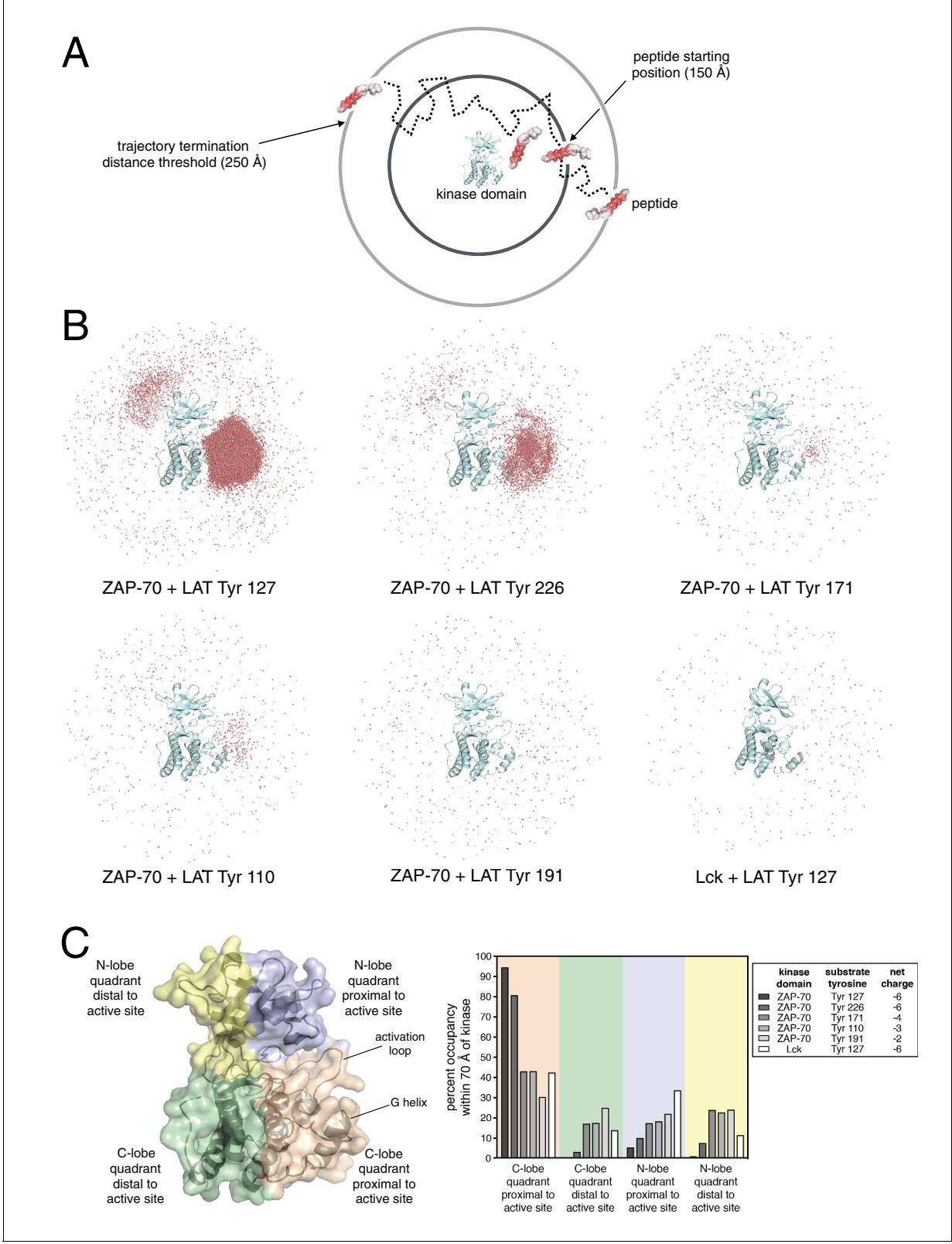

**Figure 8.** Brownian dynamics simulations of LAT peptide association with ZAP-70 and Lck kinase domains. (**A**) Schematic representation of Brownian dynamics trajectories. For each kinase-peptide pair, 10,000 trajectories were initiated by randomly placing the peptide on the surface of a sphere of radius 150 Å, centered on the center of mass of the kinase domain. Trajectories were terminated when the peptide moved more than 250 Å from the center of mass of the protein. (**B**) Distribution of various LAT Tyr Cα atoms (shown as light red dots) within 70 Å of the ZAP-70 or Lck kinase domain

*Figure 8 continued on next page*

*Figure 8 continued*

center of mass. For visualization purposes, a random sampling of 0.1% of all peptide positions within the 70 Å sphere in all 10,000 trajectories is displayed. The density of LAT Tyr Cα atoms in each panel reflects the amount of time that peptide spent in the vicinity of the kinase domain. (**C**) Frequency of substrate tyrosine Cα atoms within four quadrants encompassing the ZAP-70 or Lck kinase domain. Only peptide positions within 70 Å of the center of mass of the kinase domain, as shown in panel B, were considered for this analysis. Quadrants are defined based on atoms in the N-lobe or C-lobe of the kinase, proximal or distal to the active site, as shown in the left panel.

The following figure supplement is available for figure 8:

**Figure supplement 1.** Ionic strength-dependent phosphorylation of LAT Tyr 127 by ZAP-70.

active site (*Figure 8B*, bottom middle panel). A quantitative analysis of this feature is shown in *Figure 8C*. Tyr 127, which is a good substrate for ZAP-70 and is also phosphorylated by Lck, does not cluster around the Lck active site (*Figure 8B*, bottom right panel). Thus, long-range electrostatic steering appears to be a feature of the ZAP-70-LAT interaction that is not shared by Lck. The importance of electrostatics for substrate recruitment by ZAP-70 is substantiated by the fact that ZAP-70 activity against LAT peptides is strongly dependent on ionic strength (*Figure 8—figure supplement 1*). The role of electrostatics in substrate recognition by ZAP-70 contrasts with the previously appreciated role for electrostatics in controlling the activation of Src-family kinases, which is also salt dependent (*Ozkirimli et al., 2008*).

## A model for tyrosine kinase activation loop phosphorylation

Src-family kinases, including Lck, are activated by *trans*-autophosphorylation of their activation loops (*Cooper and MacAuley, 1988*; *Hui and Vale, 2014*; *Moarefi et al., 1997*). ZAP-70, by contrast, requires Lck or another Src-family kinase for activation (*Chu et al., 1996*; *Williams et al., 1998*), as it cannot efficiently phosphorylate its own activation loop or SH2-kinase linker (*Yan et al., 2013*) (and data not shown). The molecular basis for activation loop phosphorylation in tyrosine kinases is still not completely understood. Early studies on Src-family kinases demonstrated that they could phosphorylate peptides based on their activation loop sequences, and that the primary sequence of the peptide impacted phosphorylation efficiency; however, activation loop phosphorylation was substantially faster with full-length protein substrates when compared to peptides, indicating a role for the tertiary structure of the substrate in autophosphorylation reactions (*Casnellie et al., 1982*; *Hunter, 1982*).

We carried out a comprehensive manual inspection of all crystal structures of tyrosine kinases in the Protein Data Bank to identify plausible autophosphorylation complexes that might be suggested by crystal lattice packing. We identified one crystal structure of the insulin-like growth factor-1 receptor (IGF1-R) kinase domain in which two of the four molecules in the asymmetric unit interact in a manner that appears compatible with activation loop phosphorylation, with one kinase taking the role of the enzyme and the other that of the substrate (PDB code 3LVP) (*Nemecek et al., 2010*). Recently, this same autophosphorylation complex was identified independently in an automated survey of the Protein Data Bank (*Xu et al., 2015*). We built homology models for Lck and c-Src autophosphorylation complexes based on the IGF1-R structure (PDB code 3LVP).

Several other *trans*-autophosphorylation complexes have been proposed for tyrosine kinases, including potential structures for activation loop phosphorylation (*Wu et al., 2008*; *Xu et al., 2015*) and C-terminal tail phosphorylation (*Chen et al., 2008*). We considered these alternative proposals for *trans*-autophosphorylation complexes but favored the model based on PDB code 3LVP as it was consistent with our mutagenesis experiments (noted below, in this section and the following one). *Trans*-autophosphorylation complexes have also recently been defined for certain serine/threonine kinases (*Oliver et al., 2006*; *Pike et al., 2008*; *Zorba et al., 2014*). These structures do not provide suitable models for Lck and ZAP-70, because the shorter activation loops of the tyrosine kinases do not allow them to form the corresponding complexes.

In the Lck and c-Src models based on PDB code 3LVP, there were no clashes between the two kinase domains, and there were salt bridges at the interface that could plausibly stabilize it. The model for the Lck autophosphorylation complex is illustrated in *Figure 9A*, with one view showing the enzyme-kinase in a near-standard orientation and the other with the substrate-kinase in a similar

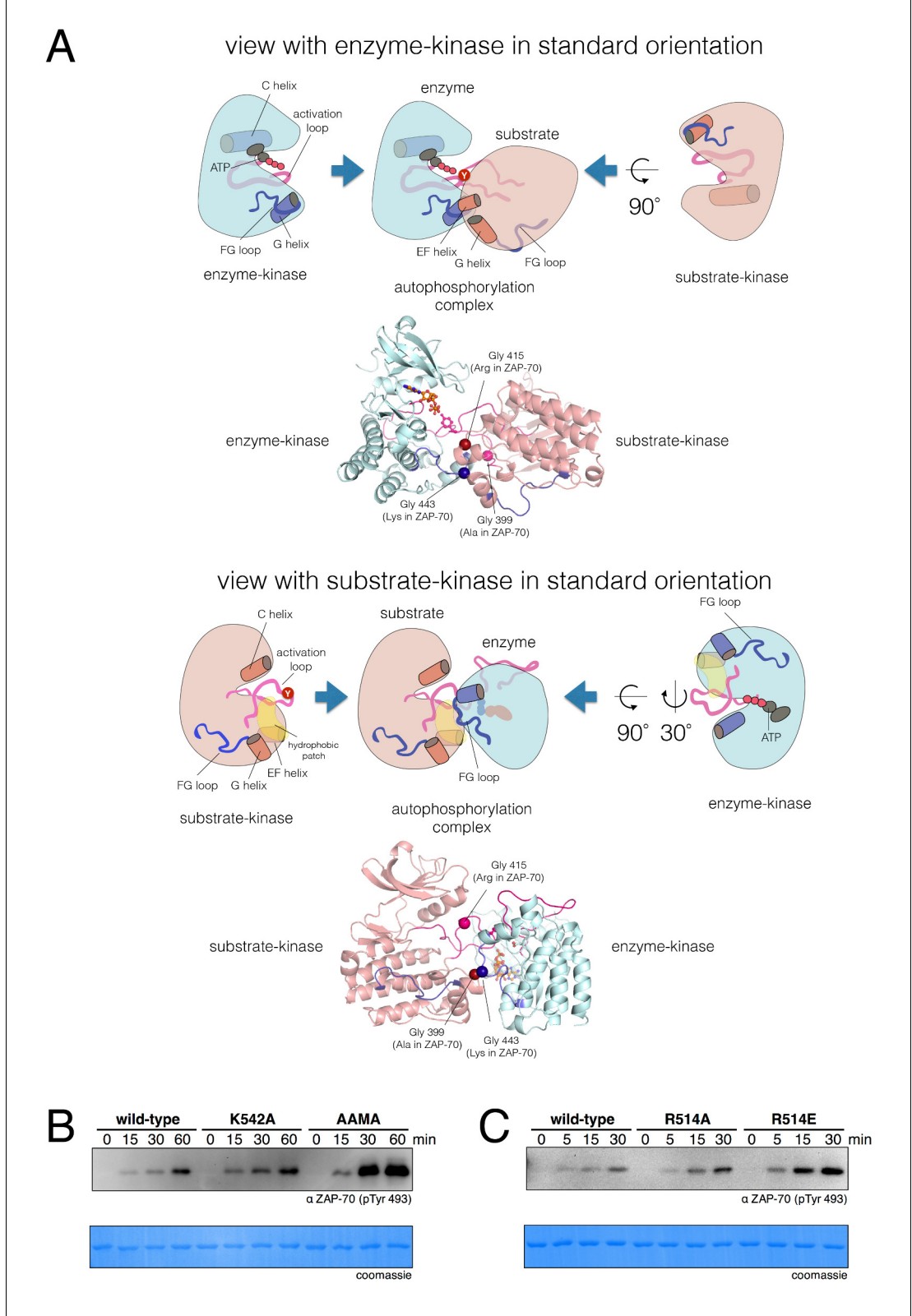

**Figure 9.** A structural model for activation loop phosphorylation of Lck. (**A**) Schematic cartoons and structural renderings of a model for Lck activation loop *trans*-autophosphorylation. This model is based on a crystal structure of IGF1-R (PDB code 3LVP) (*Nemecek et al., 2010*), in which one kinase domain molecule in the asymmetric unit, deemed the 'substrate-kinase', presents its activation loop tyrosine into the active site of a second kinase domain molecule in the asymmetric unit, deemed the 'enzyme-kinase'. (**B**) Western blot analysis of in vitro activation loop autophosphorylation

*Figure 9 continued on next page*

*Figure 9 continued*

reactions with the wild-type ZAP-70 kinase domain and FG loop mutants bearing the K542A substitution alone or simultaneous K541A, K542A, and K544A substitutions (the 'KKMK' to 'AAMA' mutant). (**C**) Western blot analysis of in vitro activation loop autophosphorylation reactions with the wild-type ZAP-70 kinase domain and mutants in which Arg 514 was substituted with alanine or glutamate. Kinase domains were used at a concentration of 5 µM in the reactions shown in panels B and C.

The following figure supplements are available for figure 9:

**Figure supplement 1.** Activation loop autophosphorylation kinetics for the wild-type Lck kinase domain (residues 229–509) and a mutant with the E448Q substitution.

**Figure supplement 2.** Activation loop autophosphorylation kinetics for the Lck and c-Src kinase domains containing substitutions in the FG loop.

orientation. In this model, as is also seen for peptide recognition by kinases, there is a short two-stranded antiparallel β sheet formed by the activation loop of the enzyme-kinase and the substrate. An obvious feature of the Lck and c-Src autophosphorylation models is that the FG loop of the enzyme-kinase is at the heart of the docking site of the substrate kinase (*Figure 9A*, top panels). The FG loop docks on a hydrophobic patch on the substrate-kinase (*Figure 9A*, bottom panels). This hydrophobic patch is formed between the G and EF helices of the substrate kinase, a known docking site on kinases (*Depetris et al., 2005*). This close packing is possible because the FG loop of the Src-family kinases, like that of IGF1-R, has small hydrophobic residues facing outwards.

To evaluate if the proposed interface is energetically stable for Src-family kinases, we initiated three molecular dynamics trajectories from this model for Lck. The simulations ranged in length from 30 ns to 120 ns, and the dimer interface remained intact in all of the trajectories. We mutated Glu 448 in Lck, which is involved in a potential intermolecular salt bridge, to glutamine, and observed a concentration-dependent loss of autophosphorylation activity in vitro (*Figure 9—figure supplement 1*). Thus, we consider the IGF1-R structure to provide a reasonable model for the enzyme-substrate docking complex that facilitates Lck activation loop autophosphorylation.

## The determinants of ZAP-70 substrate specificity also inhibit autophosphorylation

The FG loop of ZAP-70 bears four lysine residues that are important for determining selectivity for residues upstream of the tyrosine residue in substrates (*Figure 7*). This has an obvious consequence for *trans*-autophosphorylation of the activation loop. The highly charged FG loop in the enzyme-kinase is incompatible with docking on the hydrophobic patch in the substrate-enzyme in the models for the autophosphorylation complex. Additionally, in the models for the autophosphorylation complexes of Lck and c-Src, close contact between the enzyme-kinase and the substrate-kinase is enabled by the presence of a glycine residue in the FG loop (Gly 443 in Lck) of the enzyme-kinase (*Figure 9A*). Gly 443 in Lck is conserved in many tyrosine kinases, including all other Src-family kinases as well as IGF-1R, which our model is based on. Gly 443 in Lck is replaced by Lys 542 in ZAP-70, one of the positively-charged residues in the FG loop, and Lys 542 cannot be accommodated within the autophosphorylation complex.

Two other glycine residues are also important in the substrate-kinase molecule. Gly 415 in the substrate-Lck packs close to the FG loop of the enzyme-kinase (*Figure 9A*), but this residue is replaced by Arg 514 in ZAP-70, a residue that is important for determining the specificity for negatively-charged residues downstream of the tyrosine in substrate peptides (*Figure 7*). If ZAP-70 were to take the position of a substrate-kinase, then Arg 514 would be incompatible with the autophosphorylation complex, as it would come into close contact with Lys 504 in the FG loop of the ZAP-70 molcule playing the role of the enzyme. Gly 399 in the substrate-Lck is replaced by alanine in ZAP-70, which the model predicts would also lead to clashes.

One of the striking results of the specificity screens with Lck is the absence of strong selection upstream of the tyrosine residue in peptide substrates. This weak upstream selectivity in Lck is correlated with the presence of small, nonpolar residues in the FG loop in Lck and other Src-family kinases, which our modeling predicts is necessary for the formation of the autophosphorylation complex. Thus, the Src-family kinases appear to have traded upstream selectivity in substrates for

the ability to undergo efficient autophosphorylation on the activation loop. In contrast, the ability of ZAP-70 to impose strong electrostatic selectivity upstream of the tyrosine residue appears to have negated its ability to undergo efficient autophosphorylation.

We tested whether mutations at positions that clearly impact substrate specificity in ZAP-70 would also affect autophosphorylation rates. The KKMK-to-AAMA mutation in the FG loop, which reduced LAT and SLP-76 phosphorylation rates (*Figure 7—figure supplement 7*), caused a clear increase in activation loop autophosphorylation of the ZAP-70 kinase domain (*Figure 9B*). Consistent with this observation, introduction of lysine residues at corresponding positions into the Lck and c-Src kinase domains reduced their autophosphorylation rates (*Figure 9—figure supplement 2*). We observed a similar enhancement of ZAP-70 activation loop autophosphorylation when Arg 514 was replaced with alanine or glutamate (*Figure 9C*).

## Activation and substrate recognition of Lck, ZAP-70, and Syk in a model cell line

In order to connect our in vitro data to a cellular context, we studied the activities of full-length Lck and ZAP-70 in a reconstituted cell-based system in which human embryonic kidney (HEK) 293 cells were transiently transfected with various constructs of Lck or ZAP-70, together with the full-length substrates LAT or TCRζ, as described previously (*Brdicka et al., 2005*). In this system, Lck shows substantial tyrosine phosphorylation, whereas ZAP-70 is phosphorylated only when it is co-expressed with Lck (*Figure 10A*). LAT phosphorylation is only seen when Lck is co-expressed with ZAP-70. Likewise, robust TCRζ phosphorylation is only observed when Lck is present.

We also tested the importance of the adapter domains of Lck and ZAP-70 in determining specificity for LAT and the TCRζ. To do this, we made two chimeric constructs. One (Lck/ZAP-70) has the SH3 and SH2 domains of Lck fused to the kinase domain of ZAP-70. The other (ZAP-70/Lck) has the tandem SH2 domains of ZAP-70 fused to the kinase domain of Lck. The behavior of these constructs is similar to that of the wild-type protein from which the kinase domain is derived, demonstrating that the principal determinant of phosphorylation specificity is the kinase domain (*Figure 10A*).

We also analyzed the ZAP-70 paralog, Syk, as this kinase has previously been shown to have intermediate substrate specificity and auto-activation capability between ZAP-70 and Lck (*Mukherjee et al., 2013*; *Tsang et al., 2008*). In these experiments, Syk was phosphorylated in the absence of Lck, unlike ZAP-70, and it phosphorylated both LAT and TCRζ, unlike ZAP-70 and Lck.

We assessed the impact of mutations in the FG loop on the activities of Lck and ZAP-70 in HEK293 cells (*Figure 10B*). Simultaneous mutation of Lys 541, Lys 542, and Lys 544 in ZAP-70 to alanine (KKMK-to-AAMA mutant, denoted ZAP-70*) resulted in a reduction of LAT phosphorylation and an increase in TCRζ phosphorylation, even in the absence of Lck, suggesting that this mutant ZAP-70 can auto-activate. Conversely, mutation of the FG loop in Lck (PGMT-to-KKMK mutant, denoted Lck*) resulted in a loss of both Lck and TCRζ phosphorylation. No gain in LAT phosphorylation was observed for this protein.

The sequence of the FG loop in Syk resembles that of ZAP-70 but contains one glycine residue that is important for autophosphorylation in Lck (Gly 443 in Lck, Gly 575 in Syk, and Lys 542 in ZAP-70). Mutation of the FG loop sequence of Syk to that of ZAP-70 (RGMK-to-KKMK) decreased Syk activation loop autophosphorylation and also decreased the ability of Syk to phosphorylate TCRζ and LAT (*Figure 10C* and *Figure 10—figure supplement 1*). These mutations also made Syk more dependent on activation by Lck. Mutation of the FG loop sequence of ZAP-70 to that of Syk (KKMK-to-RGMK) did not result in robust autophosphorylation, presumably because Arg 514 in ZAP-70, which is a tyrosine in Syk, inhibits formation of the enzyme-substrate complex (*Figure 9C*). Similar to Syk, however, this ZAP-70 mutant could phosphorylate both LAT and TCRζ. These cell-based data are consistent with the ability of Syk to facilitate T cell receptor signaling in the absence of Lck (*Chu et al., 1996*; *Williams et al., 1998*).

## ZAP-70 and its substrates have unique sequence features

We asked if the specificity-defining sequence features of ZAP-70, LAT, and SLP-76 are unique among tyrosine kinases and their substrates. To address this question, we analyzed a curated list of experimentally-determined substrates for human tyrosine kinases from the PhosphoSitePlus database (*Hornbeck et al., 2015*). This dataset comprises roughly 1100 phosphosites for approximately

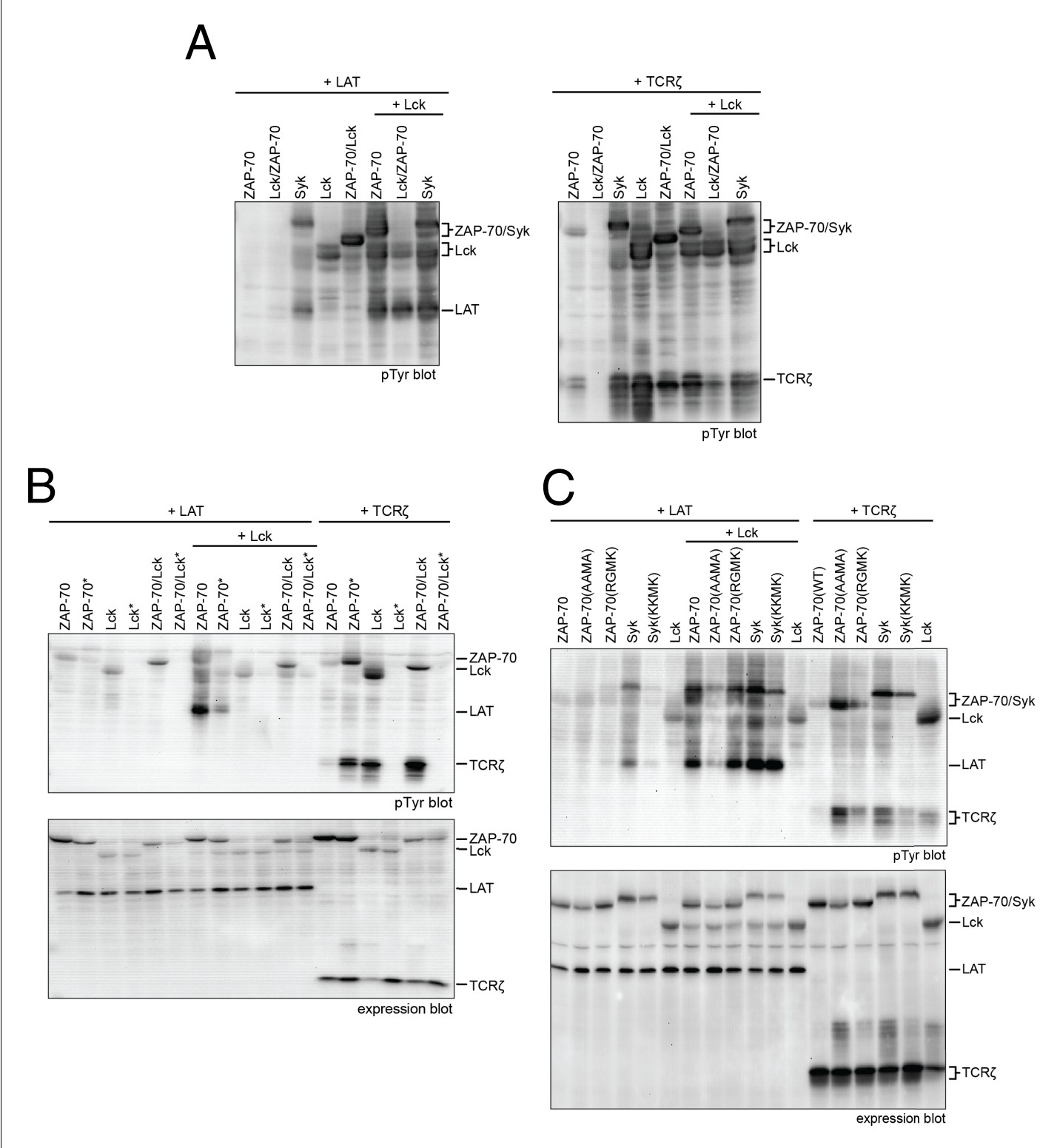

**Figure 10.** Lck, ZAP-70, and Syk activation and specificity in a model cell line. (**A**) Co-expression of wild-type full-length kinases and chimeras of Lck and ZAP-70 with LAT or TCRζ. Two chimeric kinases were used: Lck/ZAP-70 refers to a construct with the Lck SH3-SH2 module fused to the ZAP-70 kinase domain. ZAP-70/Lck refers to the ZAP-70 tandem SH2 module fused to the Lck kinase domain. (**B**) Comparison of F-G loop mutations of ZAP-70 and Lck. ZAP-70* refers to the KKMK-to-AAMA mutant, and Lck* refers to the PGMT-to-KKMK mutant. (**C**) Comparison of F-G loop mutations in ZAP-70 and

*Figure 10 continued on next page*

*Figure 10 continued*

Syk. All experiments were carried out in human embryonic kidney (HEK) 293T cells as previously described (*Brdicka et al., 2005*), and interpretation of the results is also based on the analysis in *Brdicka et al. 2005*.

The following figure supplement is available for figure 10:

**Figure supplement 1.** Western blot analysis ZAP-70 and Syk phosphorylation in HEK293 cells with site-specific antibodies.

100 tyrosine kinases. We binned these sequences based on the net charge of the sequence surrounding the tyrosine and the identity of the −1 position with respect to the phosphosite, two key determinants of ZAP-70 specificity. On average, tyrosine kinase substrates have a near-neutral net charge in proximity to the phospho-acceptor tyrosine, and the residues at the −1 position segregate into three groups: bulky-hydrophobic, polar-neutral, and negatively-charged (*Figure 11A*). Lck substrates, such as the T cell receptor ITAM phosphosites, fall into each of these major categories and are thus representative of typical tyrosine kinase substrates. LAT and SLP-76 phosphosites, however, are distinctive in having both a −1 acidic residue and a substantially negative net charge. Both of these features are conserved in LAT sequences across different vertebrates (*Figure 11B*).

ZAP-70 accommodates the unique features of LAT and SLP-76 by using a positively-charged region on its kinase domain to engage substrates. We assessed sequence conservation in this region and found that all of the lysine and arginine residues are conserved in ZAP-70 orthologs from fish to mammals. Most of these residues are not conserved in other human non-receptor tyrosine kinases (*Figure 11C*). The predicted isoelectric point of the ZAP-70 kinase domain is ~9, and it has a net charge of +7. There are three other human non-receptor tyrosine kinase domains (Fer, Tnk1, and Ctk) that have similar net charge (*Figure 11D*), but the positively-charged residues in these kinases do not cluster near the substrate binding region. Thus, the degree and placement of positive charge on its kinase domain makes ZAP-70 an outlier among tyrosine kinases.

## Concluding remarks

Our results provide a molecular explanation for the order of initial phosphorylation events during T cell receptor signaling. This sequence of events is largely dictated by the substrate specificities of the ZAP-70 and Lck kinase domains, which we have defined using a high-throughput assay to measure phosphorylation levels for hundreds of mutant peptides based on natural substrates of these kinases. Unlike Lck, and probably most other tyrosine kinases, ZAP-70 preferentially phosphorylates substrates that are enriched in negatively-charged residues, depleted of positively-charged residues, and have an aspartate at the −1 position relative to the phospho-acceptor tyrosine. These features are characteristic of the targets of ZAP-70 in LAT and SLP-76. In accommodating LAT and SLP-76, ZAP-70 not only became a highly specialized kinase but also lost the ability to activate itself by *trans*-autophosphorylation. This coupling between substrate recognition and autophosphorylation in ZAP-70, along with the electrostatic features of LAT and SLP-76, insulate these proteins from undesirable cross-talk with other signaling molecules (*Figure 12*).

A critical aspect of this electrostatic selection mechanism is that it provides stringent control over kinase-substrate interactions without restricting the diversity of downstream signals. The evolution of ZAP-70 substrates has led to the incorporation of substantial negative charge both upstream and downstream of their phospho-acceptor tyrosines. The resulting electrostatic selection is sufficiently discriminatory to ensure exclusive phosphorylation of these sites in LAT and SLP-76 by ZAP-70, while allowing each of these phosphosites to have unique functional properties. Not only can each phosphosite on LAT and SLP-76 bind to distinct SH2 domain-containing proteins (*Houtman et al., 2004*), but the rate of phosphorylation at these sites can also be tuned without compromising kinase specificity, as described earlier for LAT Tyr 132.

What evolutionary forces might have shaped the origin of the electrostatic features in LAT and SLP-76? They probably evolved not only through positive selection for efficient phosphorylation by ZAP-70, but also through negative selection against phosphorylation by other tyrosine kinases, including Lck. This type of system-wide negative selection has been noted previously for SH3

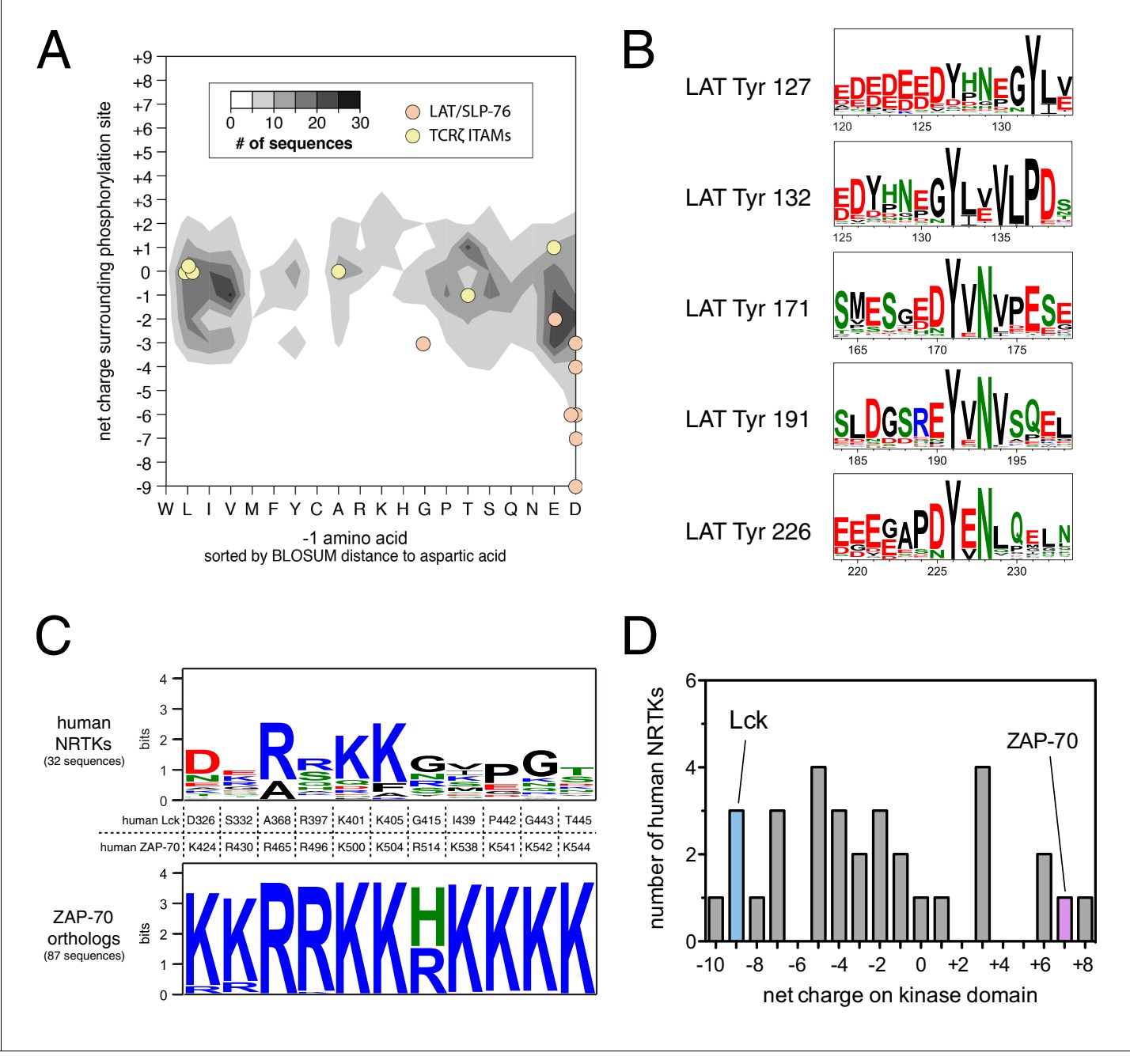

**Figure 11.** Unique sequence features of ZAP-70 and its substrates. (A) Analysis of net charge and −1 residue distributions for 1100 tyrosine kinase substrates from the PhosphoSite Plus database (*Hornbeck et al., 2015*). Analyzed sequences span seven residues on either side of the phospho-acceptor tyrosine. Data are represented as a contour plot, which depicts the prevalence of phosphosites in the database with particular combinations of −1 residue identity and net charge. Darker regions of the contour plot indicate that more sequences have that particular combination of properties. (B) Sequence logos showing the conservation around five LAT tyrosines in 59 species from fish to mammals. (C) Conservation of lysine and arginine residues important for ZAP-70 substrate recognition. The top panel displays a sequence logo showing a lack of conservation at many of these positions in human non-receptor tyrosine kinases (NRTKs). The bottom panel displays high conservation at these positions between ZAP-70 orthologs from fish to mammals. (D) Distribution of net charge for the kinase domains of all 32 human non-receptor tyrosine kinases.

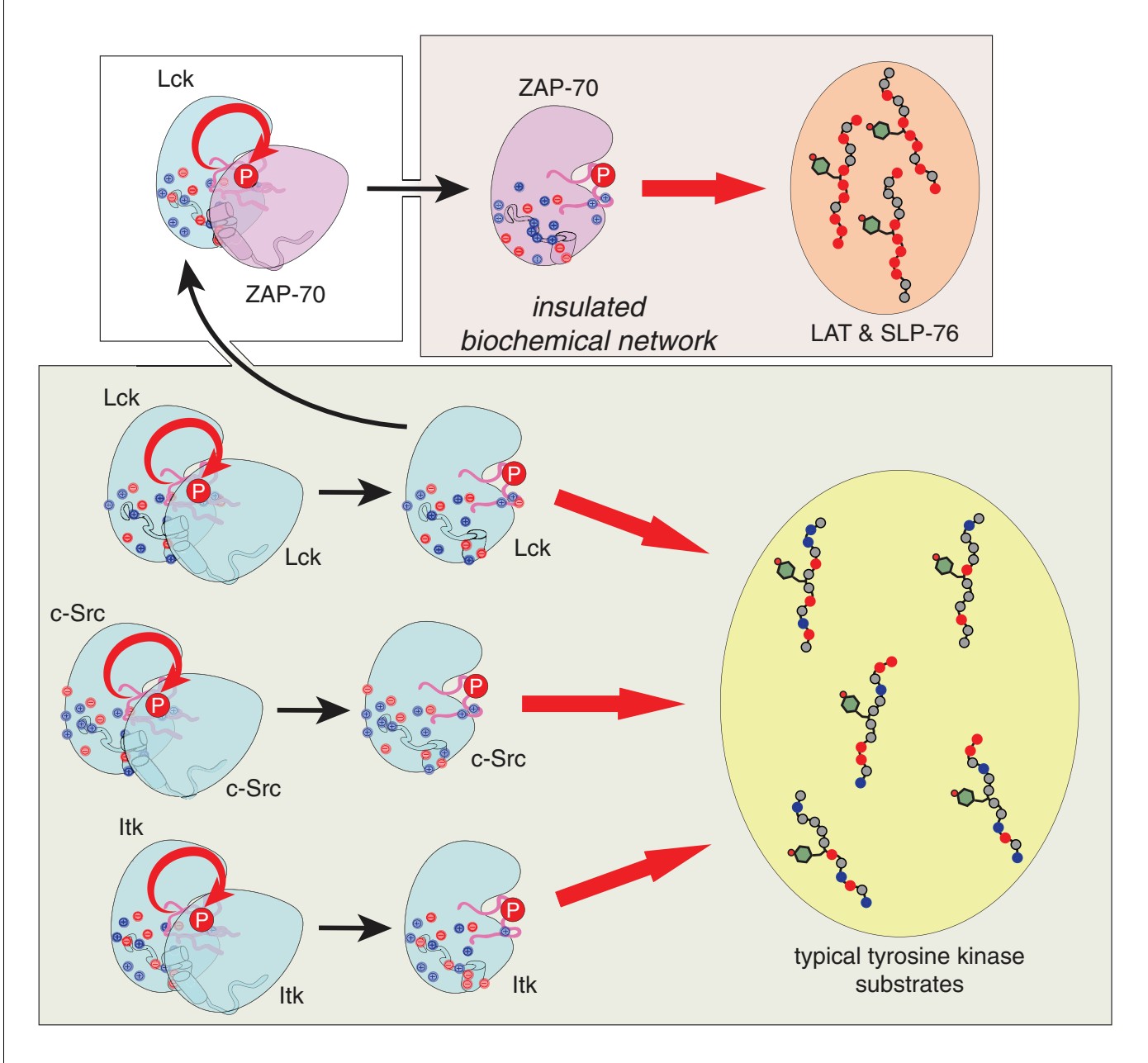

**Figure 12.** Schematic depiction of the biochemical insulation of ZAP-70, LAT, and SLP-76. Many tyrosine kinases, such as Lck, c-Src, and Itk, can activate themselves through *trans*-autophosphorylation of their activation loops (bottom panel, left side). These kinases can efficiently phosphorylate typical tyrosine kinase substrates, which have a near-neutral net charge (bottom panel, right side). ZAP-70 activation requires phosphorylation of its activation loop by Lck (top left panel). Once activated, ZAP-70 exclusively phosphorylates LAT and SLP-76, but not typical tyrosine kinase substrates, and LAT and SLP-76 cannot be phosphorylated by kinases like Lck, c-Src, and Itk (top right panel). Thus, ZAP-70 and its substrates are insulated from other proteins in the T cell receptor signaling pathway.

domain-ligand interactions in yeast, which are intrinsically promiscuous, but can achieve a remarkable degree of specificity in the cellular context (*Zarrinpar et al., 2003*).

In the case of T cell receptor-proximal kinase signaling, negative selection was presumably crucial for establishing an insulated network in which Src-family kinases cannot phosphorylate LAT. Several studies have shown that there is an active pool of Lck in the T cell, even prior to receptor stimulation (*Nika et al., 2010*; *Schoenborn et al., 2011*), and the amount of active Lck establishes a threshold

for responses to peptide ligands (*Manz et al., 2015*). Both Lck and LAT are membrane-bound, and they may cluster in the same membrane micro-domains (*Zhang et al., 1998*). Given this, if Lck could readily phosphorylate LAT, aberrant downstream signaling would result without T cell receptor stimulation by an appropriate antigen.

The tight regulation of ZAP-70 and its narrow substrate specificity are also important for faithful T cell receptor signaling, as they favor a T cell response to peptide antigens bound to MHC molecules. The MHC proteins on antigen-presenting cells are recognized by the CD4 and CD8 co-receptors, and Lck is tethered to these co-receptors at the membrane (*Figure 1A*). Upon peptide-MHC interaction with the T cell receptor and its co-receptors, Lck is repositioned near its favored substrates, the ITAMs within the cytoplasmic tails of the T cell receptor complex. This ensures not only the appropriate Lck-mediated phosphorylation of ITAMs but also the proper recruitment, phosphorylation, and activation of ZAP-70 by Lck. The strict dependence on Lck for ZAP-70 membrane recruitment and activation dictates that receptor triggering only occurs when peptide antigens are presented by MHC molecules.

B cells have an analogous pathway to the T cell receptor pathway that utilizes the Src-family kinase Lyn, the ZAP-70 paralog Syk, and the scaffold protein BLNK (or SLP-65). Unlike T cells, however, receptor activation in B cells does not require that antigens are presented on MHC molecules. Furthermore, B cell activation can be considered more flexible, because Syk can slowly auto-activate (*Tsang et al., 2008*; and it can phosphorylate ITAMs (*Chu et al., 1996*; *Mukherjee et al., 2013*), thereby making the Src-family kinase somewhat expendable (*Takata et al., 1994*). Src-family kinase participation in B cell signaling serves to speed up and increase the sensitivity of the response to B cell receptor stimulation (*Mukherjee et al., 2013*). These subtle differences between T and B cells underscore how signaling pathways and their corresponding molecules have evolved to achieve varying degrees of control. The techniques and findings we have presented here provide a framework to characterize the evolution and specialization of T and B cell kinases and to enhance our understanding of the design principles underlying lymphocyte signal transduction pathways.

## Materials and methods

### Preparation of DNA libraries for high-throughput specificity screens

LAT/SLP-76/p38α/TCRζ library

A pBAD33 vector containing the enhanced circularly permuted OmpX protein (eCPX) (*Rice and Daugherty, 2008*) was purchased from Addgene. This construct was modified to insert a Strep-tag (encoding the peptide sequence WSHPQFEK) downstream of the eCPX gene using QuikChange site-directed mutagenesis. DNA encoding the peptides listed in *Figure 3—figure supplement 1*, were inserted in between the signal sequence DNA and the 5' end of the eCPX gene as follows. Two oligonucleotide primers were designed to tile the peptide gene with roughly 20 overlapping bases. The 'outer primer' contained a sequence bearing an SfiI restriction site followed by the first ~2/3 of the DNA sequence of the peptide. The 'inner primer' contained the last ~2/3 of the peptide DNA, followed by a sequence that was homologous to a linker sequence before the eCPX gene. A third primer was designed to anneal downstream of the Strep-tag, encompassing a second SfiI site, and read in the reverse direction.

DNA encoding the peptide-eCPX-Strep-tag fusion was amplified in two steps by PCR. In the first step, the inner primer and the reverse primer were used with the eCPX-Strep-tag plasmid as the template. The product of this reaction was used as a template for a second PCR reaction with the outer primer and the reverse primer to yield the full peptide-eCPX-Strep-tag gene fusion. This linear DNA was digested with SfiI and ligated into an SfiI-digested pBAD33-eCPX vector to yield the desired plasmid. This procedure was carried out individually for all of the peptides in this library, and the sequence of each plasmid was verified by Sanger sequencing. Concentrations of each plasmid were determined based on their absorbance at 260 nm, and the plasmids were mixed at an equimolar ratio to yield the final library.

Scanning point-mutagenesis libraries

Scanning point mutant libraries of peptides were constructed as previously described for the comprehensive point-mutagenesis of a small protein domain (*McLaughlin et al., 2012*). For this

approach, sub-libraries of DNA inserts were created in which all possible amino acid substitutions at a single position on the peptide were encoded using synthetic oligonucleotides bearing a degenerate codon at the desired position (NNS, where N refers to A, T, G, or C, and S refers to G or C). Each sub-library comprised a mixture of 32 different peptide-eCPX-Strep-tag gene fusions that encoded all of the possible point mutants at one position on the peptide. These inserts were amplified using the same two-step PCR protocol described above for the LAT/SLP-76/p38α/TCRζ library, but with the appropriate NNS codon in the inner and outer primers. One sub-library insert was produced for each position in the peptide, so for a 20 residue peptide, 20 inserts were produced.

All of the sub-library inserts were independently gel-purified. The concentrations of these purified fragments were determined using the PicoGreen reagent and then pooled with equal stoichiometry. The pooled inserts were digested with SfiI and ligated into an SfiI-digested pBAD33-eCPX vector using T4 DNA ligase. The ligation mixture was purified and concentrated using a spin column, eluted in water, and used to transform TOP10 cells by electroporation. The transformed cells were grown overnight at 37°C in a selective liquid medium containing 25 μg/mL of chloramphenicol. The following day, the DNA was isolated by mini-prep to yield the final library. The distribution of each variant in the libraries was assessed by Illumina deep sequencing. Typically, DNA corresponding to all of the point mutants was present in roughly equal numbers (within a factor of 3–4), and the wild-type sequence was over-represented approximately 20-fold, as expected based on the cloning strategy.

## High-throughput specificity screens
### Surface-display of the peptide libraries
Bacteria bearing the surface-displayed peptide libraries were prepared using previously described protocols with slight modifications (*Henriques et al., 2013*; *Rice and Daugherty, 2008*). Typically, 100 ng of library DNA was used to transform 100 μL of electrocompetent *E. coli* MC1061 cells. After electroporation, the cells were resuspended in 1 mL of LB and allowed to recover at 37°C for 1 hr. 100 μL of the recovered cells were used to inoculate 5 to 10 mL of LB with 25 μg/mL of chloramphenicol, and this liquid culture was incubated overnight at 37°C.

100 μL of the dense overnight cell culture was used to inoculate 5 mL of LB with 25 μg/mL of chloramphenicol. These cells were grown at 37°C for 3 to 4 hr until they reached an optical density at 600 nm ($OD_{600}$) of 0.5. Expression of the eCPX scaffold was induced by adding arabinose to a final concentration of 0.4% (w/v), and the cells were incubated at 25°C for an additional 4 hr. After induction, the $OD_{600}$ was typically between 0.9 and 1.0. Small aliquots of the cells (150 to 450 μL) were transferred to microcentrifuge tubes and centrifuged at 4000 rcf and 4°C for 15 min. After removing the medium in the supernatant, the cells were resuspended in phosphate buffered saline (PBS) and centrifuged again. The PBS was removed and the cell pellets were stored overnight at 4°C.

### Phosphorylation of cells and fluorescence-activated cell sorting
Cells were prepared for phosphorylation by resuspending them in a buffer containing 50 mM Tris, pH 7.5, 150 mM NaCl, 5 mM $MgCl_2$, 1 mM TCEP, and either the ZAP-70 or Lck kinase domain at a concentration ranging from 100 nM to 1 μM. The cell suspensions, typically with an $OD_{600}$ between 1.0 and 1.5, were warmed on a 37°C heat block. Reactions were initiated by addition of ATP to a final concentration of 1 mM. At the desired time point, 100 μL of the suspension was transferred to a cold microcentrifuge tube, and EDTA was added to a final concentration of 25 mM to quench kinase activity (specific kinase concentrations and reaction times are given in the table below). The quenched cell suspensions were centrifuged at 4000 rcf for 15 min and washed once with PBS. To label the phosphorylated cells, the pellets were resuspended in 100 μL of Millipore 4G10 PE conjugate (1:50 dilution) or 100 μL of Cell Signaling Technology P-Tyr-100 PE conjugate (1:50 dilution), diluted in PBS containing 0.2% bovine serum albumin (PBS+BSA). The cells were incubated with the antibody for 1.5 hr on ice in the dark, then centrifuged and washed once with PBS+BSA.

The labeled and washed cells were resuspended in 1 mL of PBS+BSA then diluted 10-to-15-fold in PBS+BSA for sorting. Cells were sorted on a BD Influx cell sorter equipped with PE and FITC filters. The distribution of PE fluorescence (phosphorylation level) for the whole cell population was measured, and a sorting gate was set to collect the highest 15–25% of this distribution. For each sample, 1,000,000 cells were collected into a 15 mL conical tube containing 3 mL of LB on ice. For

each library, an analogous tube was prepared with approximately 1,000,000 unsorted cells diluted in LB.

| Figure | Library | Kinase | [Kinase] | Reaction Time | Antibody |
|---|---|---|---|---|---|
| 3B | LAT/SLP-76/p38α/TCRζ | ZAP-70 | 400 nM | 30 min | P-Tyr-100 |
| 3B | LAT/SLP-76/p38α/TCRζ | Lck | 400 nM | 30 min | P-Tyr-100 |
| 5A | LAT Tyr 226 | ZAP-70 | 400 nM | 5 min | 4G10 |
| 5C | LAT Tyr 127 | ZAP-70 | 100 nM | 15 min | 4G10 |
| 5D | LAT Tyr 132 | ZAP-70 | 1000 nM | 15 min | 4G10 |
| 6A | TCRζ Tyr 111 | Lck | 100 nM | 5 min | P-Tyr-100 |
| 6B | TCRζ Tyr 123 | Lck | 100 nM | 15 min | P-Tyr-100 |
| 6C | LAT Tyr 127 | Lck | 100 nM | 15 min | 4G10 |

## DNA isolation and deep sequencing

The sorted cells were centrifuged at 4000 rcf for 20 min. The supernatant was decanted carefully and the tubes were inverted over a paper towel to remove excess liquid (note that the pellets were not visible). The cell pellet was resuspended with 50 µL of water by vigorous pipetting in the bottom of the conical tube. The suspension was transferred to a microcentrifuge tube and boiled for 10 min to lyse the cells. The lysate was centrifuged at 13,000 rcf, then 10 µL of the supernatant was used as the template in a 50 µL PCR reaction to amplify the peptide genes with flanking Illumina TruSeq adapter sequences. After amplification, the PCR reaction mixtures were diluted 100-fold, and this diluted solution was used as a template for a second round of PCR to append unique 5' and 3' indices for each sample. Multiple samples from different experiments were pooled and sequenced on an Illumina MiSeq system using standard protocols for paired-end sequencing. Typically, DNA was loaded to obtain at least 500 read counts for each variant in an unsorted library and equivalent total read counts for all of the variants in unsorted and sorted libraries.

## Analysis of deep sequencing data

Raw paired end reads were merged using the software PEAR with a quality score threshold of 30 (*Zhang et al., 2014*). Adapter sequences and any additional bases preceding or following the peptide genes were removed using the software Cutadapt (*Martin, 2011*). The read counts for each variant in the trimmed sequence files were determined using scripts written in Biopython (*Cock et al., 2009*). The frequency ($f_x$) of each variant $x$ in a sample was expressed as the ratio between the counts for that variant and the total counts for all variants in that sample. Normalized enrichment values were determined by correcting the frequency of a variant in a sorted population for that in an unsorted population then normalizing this corrected frequency to a reference member of the library, as previously described (*McLaughlin et al., 2012*):

$$normalized\ enrichment = \frac{\left(\frac{f_x^{sorted}}{f_x^{unsorted}}\right)}{\left(\frac{f_{reference}^{sorted}}{f_{reference}^{unsorted}}\right)}$$

For scanning point-mutagenesis screens, the normalized enrichment was typically expressed as a logarithmic value with the wild-type peptide as the reference.

$$\log_{10}(normalized\ enrichment) = \log_{10}\left(\frac{f_x^{sorted}}{f_x^{unsorted}}\right) - \log_{10}\left(\frac{f_{wild-type}^{sorted}}{f_{wild-type}^{unsorted}}\right)$$

## Peptide and protein expression and purification

### Peptide purifications

All of the peptides used for in vitro kinetic assays were generated recombinantly. Peptides were cloned into a pET vector as C-terminal fusions to a His$_6$-tagged SUMO protein. The SUMO-peptide fusions were expressed in BL21(DE3) cells by overnight induction at 18°C. Cells were pelleted and then resuspended in a buffer containing 50 mM Tris, pH 8.0, 300 mM NaCl, 10 mM imidazole, 2 mM β-mercaptoethanol, and a cocktail of protease inhibitors. Cells were lysed by sonication, and the lysates were clarified by centrifugation at 35,000 rcf for 1 hr. The SUMO-peptide fusions were separated from contaminants in the supernatant by binding to Ni-NTA resin followed by extensive washes in lysis buffer. The fusion protein was eluted with a buffer containing 50 mM Tris, pH 8.0, 300 mM NaCl, 250 mM imidazole, and 2 mM β-mercaptoethanol. The eluted protein was concentrated and then treated with the reducing agent TCEP (5 mM) and the SUMO-specific protease Ulp1 overnight at room temperature. The cleaved peptide was separated from Ulp1 and SUMO by reverse phase high performance liquid chromatography (RP-HPLC) on an Agilent 1200 series HPLC, using a semi-preparative C$_{18}$ column and a water-to-acetonitrile gradient with 0.1% trifluoroacetic acid. The identity of each peptide was confirmed by electrospray ionization mass spectrometry, and the purity of each peptide (>95%) was confirmed by RP-HPLC on an analytical C$_{18}$ column. Typical yields using this protocol ranged from 5 to 20 mg of purified peptide per 1 liter of bacterial expression culture.

### Protein purifications

With the exception of the c-Src kinase domain constructs, which were expressed and purified using established methods (*Seeliger et al., 2005*), all proteins were generated by a baculovirus expression system in *Spodoptera frugiperda* 21 (Sf21) cells using standard protocols. The human ZAP-70 and Lck constructs (and mutants, thereof) were cloned into a pFastBac1 plasmid. The ZAP-70 kinase domain construct contained residues 327–606 followed by a C-terminal His$_6$-tag, and the Lck kinase domain construct contained residues 229–509 with a N-terminal His$_6$-tag and TEV protease site.

For a typical purification, 2 L of Sf21 cells were infected with the baculovirus encoding the desired protein and grown for two days. After removal of the medium by centrifugation, the cells were resuspended in a lysis buffer containing 50 mM Tris, pH 8.0, 50 mM NaCl, 10 mM imidazole, 2 mM β-mercaptoethanol, and 10% glycerol. The suspension was supplemented with 1 mM MgCl$_2$, 10 μg/mL DNase, and a protease inhibitor cocktail. Cells were lysed using an Avestin EmulsiFlex C50 homogenizer, and the lysate was clarified by ultracentrifugation to remove insoluble debris. The supernatant was applied to a Ni-NTA column, and the flow-through was collected. The resin was washed with increasing concentrations of NaCl up to 300 mM, followed by increasing concentrations of imidazole up to 250 mM to elute the protein. The eluted kinase solution was flowed over a GE HiPrep 26/10 to reduce the salt concentration to 50 mM and remove the imidazole.

Each protein was further purified by ion exchange. For the ZAP-70 kinase domain constructs, the protein solution was applied to a HiTrap Q anion exchange column in a Tris buffer at pH 7.0, and the kinase was collected in the flow-through. For the Lck constructs, a HiTrap Q column was also used at pH 7.0, however the protein bound to the column and was eluted over a salt gradient from 50–500 mM NaCl. The His$_6$-tags were not removed from the kinase domain constructs. All of the proteins were purified by size exclusion chromatography over a Superdex 200 column and stored in a buffer containing 10 mM HEPES, pH 7.5, 150 mM NaCl, 5 mM MgCl$_2$, 1 mM TCEP, and 10% glycerol.

For the Lck kinase domain constructs and certain ZAP-70 kinase domain mutants, the proteins were isolated from Sf21 cells bearing phosphotyrosine modifications, as determined by western blot and/or an SDS-PAGE gel-shift. This phosphorylation was readily reversed by treatment of the proteins with the non-specific tyrosine phosphatase YopH. Thus, to purify homogeneously unphosphorylated proteins, kinases were either treated with YopH after the initial nickel column purification step, before ion exchange, or they were treated with YopH after the complete purification protocol and then re-purified over a Ni column followed by size exclusion chromatography.

## Biochemical assays with purified kinases and peptides

### Continuous colorimetric assay for kinetic measurements of kinase activity

We used a continuous colorimetric assay to measure the phosphorylation of kinetics of purified peptides by purified kinases (*Barker et al., 1995*). In this assay, ADP production is coupled to NADH oxidation through two non-rate-limiting enzymatic steps, with concomitant loss of NADH absorbance at 340 nm. In all experiments, the reaction solution contained 50 mM Tris, pH 7.5, 100 mM NaCl, 10 mM MgCl$_2$, 1 mM phosphoenolpyruvate, 300 µg/mL NADH, 2 mM sodium orthovanadate, 100 µM ATP, and an excess of pyruvate kinase and lactate dehydrogenase (approximately 120 units/ mL and 80 units/mL, respectively, added from a commercially available mixture of the two enzymes). The peptide concentration in all experiments was 500 µM unless stated otherwise. Reactions were initiated by the addition of the kinase to a final concentration of 1 µM. Reaction progress was monitored by measuring absorbance at 340 nm every 10 s at 25°C on a SpectraMax plate reader. All reported rates for peptide phosphorylation correspond to the difference in the initial velocity for ADP consumption by a kinase in the presence and absence of a peptide substrate.

### Western blot analysis of kinase autophosphorylation

Kinase autophosphorylation reactions were monitored by western blot. In all reactions, the kinase was diluted to a concentration between 50 nM and 5 µM in a buffer containing 50 mM Tris, pH 7.5, 150 mM NaCl, 5 mM MgCl$_2$, 2 mM sodium orthovanadate, and 1 mM TCEP. Phosphorylation reactions were initiated by addition of ATP to a final concentration of 1 mM. Reactions were typically carried out at 25°C. At various time points, aliquots of the reaction solution were mixed with SDS-PAGE loading dye supplemented with additional EDTA to achieve final concentrations of 1% SDS and 12.5 mM EDTA in the quenched solutions. Time points were run on 12% acrylamide SDS-PAGE gels and then transferred to PVDF membrane using a semi-dry transfer apparatus. Lck samples were transferred using Towbin transfer buffer (25 mM tris, 192 mM glycine, 20% MeOH), and ZAP-70 samples were transferred using CAPS transfer buffer (10 mM CAPS, pH 11, 10% MeOH).

Membranes were blocked for one hour at room temperature with 4% dried milk powder (w/v) dissolved in Tris-buffered saline with 0.1% Tween-20 (TBST). Primary antibodies were diluted with 4% milk (w/v) in TBST and applied overnight at 4°C. The following antibodies and dilutions were used: Millipore 4G10 (05–321) mouse pTyr antibody (1:2000), Cell Signaling Technologies phospho-Src Family (Tyr 416) Antibody #2101 (1:2000), and Cell Signaling Technologies phospho-Zap-70 (Tyr 493)/Syk (Tyr 526) Antibody #2704 (1:2000). After overnight incubation with primary antibody, membranes were washed extensively with TBST. Secondary antibody conjugates to horseradish peroxidase were applied at a 1:1000 dilution in TBST with 4% milk for one hour at room temperature. Blots were washed with TBST, treated with enhanced chemiluminescence reagents, and imaged. For most experiments with high kinase concentrations, the membranes were stained with coomassie brilliant blue to visualize protein levels in each sample. For experiments with low kinase concentrations, a separate SDS-PAGE gel was run and directly stained with coomassie brilliant blue.

## Cell-based assays for kinase activation and specificity

The cell-based assays for kinase activation and specificity were carried out as described previously (*Brdicka et al., 2005*; *Yan et al., 2013*). HEK 293T cells were transiently co-transfected using Lipofectamine and Plus reagents (Invitrogen) with expression constructs for full-length LAT or the T cell receptor ζ chain along with different expression constructs for the three kinases, ZAP-70, Lck, or Syk, as indicated in *Figure 10*. Cells were lysed by resuspension in an SDS-PAGE gel loading dye, and cellular debris was removed by ultracentrifugation. The supernatants from the cell lysates were analyzed by western blotting. Phosphorylation of the protein constructs was analyzed with the Millipore 4G10 pan-phosphotyrosine antibody. Expression levels were monitored using specific antibodies for each protein.

## Computational analyses of kinase-substrate interactions

### Modeling and molecular dynamics simulations of ZAP-70 bound to a LAT-based peptide

There are no published crystal structures of substrate complexes of ZAP-70 or its close relative Syk. To characterize the substrate-binding region of ZAP-70, we first examined the structures of the

substate-free ZAP-70 kinase domain (PDB code 1U59) (*Jin et al., 2004*) and Lck kinase domain (PDB code 1QPJ) (*Zhu et al., 1999*). The electrostatic surface potential of the protein molecules in these crystal structures was calculated using the APBS software (*Baker et al., 2001*) and visualized in PyMOL (*Schrödinger, 2015*) (*Figure 7A*).

For molecular dynamics simulations of ZAP-70 bound to a LAT-based peptide, we built starting structures by modifying the ZAP-70 kinase domain crystal structure (PDB code 1U59) using the software Coot (*Emsley et al., 2010*). First, the active-site bound staurosporine, C-terminal His$_6$-tag, and activation loop atoms were removed from the coordinate file. A disordered loop in the N-lobe of the kinase was modeled into the structure using the corresponding residues from an inactive structure of full-length ZAP-70 (PDB code 4K2R) (*Yan et al., 2013*). Then, crystal structures of the substrate-bound insulin receptor tyrosine kinase (PDB codes 1IR3 and 1GAG) (*Hubbard, 1997*; *Parang et al., 2001*) were used as a template to model the ZAP-70 activation loop sequence, ATP, two Mg$^{2+}$ ions, and four substrate residues (the phosphoacceptor tyrosine and three residues immediately downstream). The remaining substrate residues were modeled in an arbitrary configuration away from the kinase domain. The ZAP-70 sequence used in this model corresponds to residues 328–602 of the human sequence with N-terminal acetyl and C-terminal carboxamide caps. The human LAT sequence used corresponds to residues 219–233, spanning Tyr 226, capped with N-terminal acetyl and C-terminal carboxamide functional groups (peptide sequence: EEEGAPDYENLGE LN).

Molecular dynamics trajectories were generated using Amber14 (*Case et al., 2014*), and the Amber ff99SB force field was used for all calculations (*Lindorff-Larsen et al., 2010*). In all simulations, the TIP3P water model was used, and no additional ions were required to neutralize the system. After the initial energy minimization steps, the system was heated to 300 K, followed by four 500 ps equilibration steps at constant number, pressure, and temperature (NPT), with harmonic positional restraints on the protein, peptide, ATP, and Mg$^{2+}$ atoms. Next, the system was subject to another 1 ns equilibration step at constant number, volume, and temperature, with no positional constraints on any atoms. Finally, production runs were carried out under NPT conditions. Periodic boundary conditions were imposed, and particle-mesh Ewald summations were used for long-range electrostatic calculations. The van der Waals cut-off was set at 10 Å. A time step of 2 fs was employed and the structures were stored every 2 ps.

## Brownian dynamics simulations

Brownian dynamics simulations were performed using the Simulation of Diffusional Association (SDA) program (*Gabdoulline and Wade, 1998*; *Martinez et al., 2015*), with the structures of the kinase domains of ZAP-70 (PDB ID 1U59) and Lck (1QPJ). A phosphorylated tyrosine residue at position 394 in Lck was replaced with an unphosphorylated tyrosine residue. The LAT peptides were built in an extended β conformation with the AMBER LEaP program (*Case et al., 2014*). The APBS program was used to compute the electrostatic potential grid around the protein and the respective peptides (*Baker et al., 2001*). The solvent dielectric constant was set at 78.0 and the dielectric constant of the solute (protein) interior was set at 2.0. All of the simulations were run at an ionic strength corresponding to 100 mM NaCl and a temperature of 298 K. The effective charges on the surface of the protein and peptides were computed with the ECM module of SDA (*Gabdoulline and Wade, 1996*).

During the simulations the kinase domain was held fixed and trajectories for the rigid peptides about the protein were computed by solving the Ermak-McCammon algorithm (*Ermak and McCammon, 1978*) using SDA. We used mutual rotation and translation diffusion coefficients of the respective molecules, calculated from their hydrodynamic radius of gyration using the Stokes-Einstein relation. Each trajectory was started by placing the peptide at random positions on the surface of a sphere centered at the center of mass of the protein, with a radius of 150 Å. Each trajectory was run until the peptide moved to a distance greater than 250 Å from the center of the protein. 10,000 trajectories were generated for each peptide.

A time step of 1 ps was used when the distance between the peptide and protein was small (less than 64–68 Å, depending on the size of the sidechains of the peptide). Beyond this distance the time step was allowed to increase up to 20 ps. In order to avoid traps, the peptide was deliberately

moved by 1 Å in an arbitrary direction if there was no change in position/orientation for over 150 steps.

## Molecular dynamics simulations of the model for the Lck autophosphorylation complex

Homology models of Lck and c-Src autophosphorylation complexes were built using the software Modeller (*Webb and Sali, 2014*), implemented in Chimera (*Pettersen et al., 2004*). A crystal structure of IGF-1R (PDB code 3LVP), which contains four protein molecules in the asymmetric unit, was used as the template (*Nemecek et al., 2010*). In this structure, chain C adopts the enzyme-kinase role and chain B adopts the substrate-kinase role. The Lck homology model was used as a starting structure for molecular dynamics simulations after docking ATP and one Mg$^{2+}$ ion into the active site of each kinase. MD simulations were set up and run analogously to those for the LAT-bound ZAP-70 simulations, except that the Gromacs 4.6.2 package was used (*Pronk et al., 2013*).

## Bioinformatic analyses

### Compilation and analysis of orthologous LAT sequences

The LAT sequences used to generate *Figure 5—figure supplement 5* and *Figure 11B* were compiled through a series of protein-protein BLAST searches in the NCBI non-redundant protein database (*Altschul et al., 1990*; *Pruitt et al., 2005*). Initially, the human LAT sequence was used as the query, and in additional rounds of searches, sequences from non-mammal vertebrates were used to find sequences from more distantly-related organisms. Subject sequences identified in the BLAST searches were kept if they met three criteria: (1) the N-terminal ~30 amino acids should be predominantly hydrophobic and followed by a 'CXXC' lipidation motif, consistent with the membrane anchor in human and mouse LAT, (2) the C-terminal ~100 amino acids should contain at least 3 or 4 tyrosine residues, and (3) the tyrosine residues should be surrounded by sequences that resemble SH2 binding sites: for example, the Y[I/L]XV motif at Tyr 132 represents a PLCγ1 binding site (*Songyang et al., 1993*). Using this approach, we identified 43 mammalian LAT sequences and 16 non-mammalian vertebrate LAT sequences. The regions surrounding each tyrosine were manually aligned with no gaps, and sequence logos were generated using the online tool WebLogo (*Crooks et al., 2004*).

### Compilation and analysis of orthologous ZAP-70 sequences and NRTK sequences

The ZAP-70 sequences used to generate *Figure 11C* were compiled through iterative BLAST searches, analogously to the LAT sequences. Two criteria were used to determine if a sequence should be retained: (1) the sequence should contain the canonical SH2-SH2-kinase domain architecture of Syk-family kinases, and (2) the sequence should have highest homology to ZAP-70, as opposed to Syk, when used as the query in a BLAST search against the human proteome. In total, 87 ZAP-70 orthologs were unambiguously identified in organisms ranging from cartilaginous fish to mammals. Sequences were aligned using the T-Coffee multiple sequence alignment program (*Notredame et al., 2000*), and the sequence logo was generated using WebLogo (*Crooks et al., 2004*).

The sequence logo for human non-receptor tyrosine kinases was generated by manually compiling all 32 kinase domain sequences followed by the same alignment and visualization protocol. Net charge on kinase domains sequences was estimated by summing the number of lysine and arginine residues and subtracting the sum of the number of aspartate and glutamate residues.

### Analysis of tyrosine kinase substrates

The analysis of tyrosine kinase sequences shown in *Figure 11A* was carried out using sequences obtained from a manually curated list of kinase-substrate pairs on the PhosphoSitePlus database (*Hornbeck et al., 2015*). As of May 14, 2016, the dataset contained over 16,000 kinase-substrate pairs from a variety of organisms. This parent list was filtered to only select human kinase-substrate pairs in which a substrate tyrosine was the reported phosphosite, resulting in 1118 non-redundant substrate sequences. Each sequence comprised 7 residues upstream and 7 residues downstream of the tyrosine phosphosite. The net charge of each sequence was estimated by summing the number

of lysine and arginine residues and subtracting the sum of the number of aspartate and glutamate residues. Sequences were sorted based on net charge and the identity of the −1 residue relative to the phosphosite. The −1 residue similarities were assessed based on the BLOSUM score relative to aspartate (*Henikoff and Henikoff, 1992*).

## Acknowledgements

We thank Hector Nolla and Alma Valeros of the Flow Cytometry Core Facility at UC Berkeley for their assistance with cell sorting; Xiaoxian Cao for assistance with insect cell expression of kinases; Pradeep Bandaru for assistance with analysis of deep sequencing data; and Jeanine Amacher, Yong-jian Huang, Yasushi Kondo, and Yamuna Krishnan for critically reading the manuscript. This work was supported in part by NIH grant PO1 AI091580 to AW and JK. NHS is supported by the Damon Run-yon Cancer Research Foundation postdoctoral fellowship. QW and QY were supported by Cancer Research Institute Irvington postdoctoral fellowships.

## Additional information

### Competing interests

JK: Senior editor, *eLife*. The other authors declare that no competing interests exist.

### Funding

| Funder | Grant reference number | Author |
| --- | --- | --- |
| National Institutes of Health | PO1 AI091580 | Arthur Weiss John Kuriyan |
| Damon Runyon Cancer Research Foundation | | Neel H Shah |
| Cancer Research Institute | | Qi Wang Qingrong Yan |

The funders had no role in study design, data collection and interpretation, or the decision to submit the work for publication.

### Author contributions

NHS, Conception and design, Acquisition of data, Analysis and interpretation of data, Drafting or revising the article; QW, QY, DK, TAK, Conception and design, Acquisition of data, Analysis and interpretation of data; IRF, WPR, Acquisition of data, Analysis and interpretation of data; RR, AW, JK, Conception and design, Analysis and interpretation of data, Drafting or revising the article

### Author ORCIDs

Arthur Weiss, http://orcid.org/0000-0002-2414-9024
John Kuriyan, http://orcid.org/0000-0002-4414-5477

## Additional files

### Supplementary files

• Supplementary file 1. Trajectory file for a representative molecular dynamics simulation of the ZAP-70 kinase domain bound to a LAT segment surrounding Tyr 226. This PDB file contains 500 instantaneous structures sampled every 1 ns from a 500 ns trajectory. The human ZAP-70 kinase domain (residues 328 to 602), ATP, and two $Mg^{2+}$ ions are defined as chain A. Human LAT residues 219 to 233 are defined as chain B. Numbering in the PDB file corresponds to the conventional numbering of the human protein sequences. Water molecules were removed from the coordinate file for clarity and to constrain the file size. These coordinates correspond to 'run 2' of the five simulations described in *Figure 7—figure supplements 2–6*.

• Supplementary file 2. Script to render MD trajectory in PyMOL. Upon loading *Supplementary file 1* into PyMOL (*Schrödinger, 2015*), running this script will render the molecules and highlight key residues similar to the images shown in *Figure 7*.

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

## Appendix

# Analysis of surface-display levels for peptide libraries

The platform we have used to derive the efficiency of phosphorylation of peptides by ZAP-70 and Lck relies on the expression of a scaffold protein (eCPX), fused to individual peptides, on the surface of *E. coli* cells. The eCPX scaffold allows us to display two peptides per scaffold on the bacterial surface (*Rice and Daugherty, 2008*). In our experiments, the N-terminus of eCPX was fused to the substrate peptide, and the C-terminus was fused to a Strep-tag (with the sequence WSHPQFEK), which we can detect using a sequence-specific antibody and thus use as a marker for surface-display level. Here, we describe the variation in surface-display levels for different wild-type parent peptides, as well as mutant peptides in a scanning mutagenesis library. We first describe the results for different parent peptides.

To measure surface-display levels of individual peptides within a library, *E. coli* cells were grown and expression of the eCPX scaffold fused to the peptides was induced as described for the phosphorylation screens. After expression, cells were labeled with a Strep-tag-specific antibody conjugated to a FITC-like dye (IBA StrepMAB-Classic Chromeo 488 conjugate). The cells were analyzed by flow cytometry and divided into 5 or 6 bins spanning the distribution of surface-display levels (*Appendix 1—figure 1*). In a typical experiment, ~$10^6$ cells were collected from each bin. DNA from each bin of cells, as well as from an unsorted population, was isolated and deep sequenced as described for the phosphorylation screens.

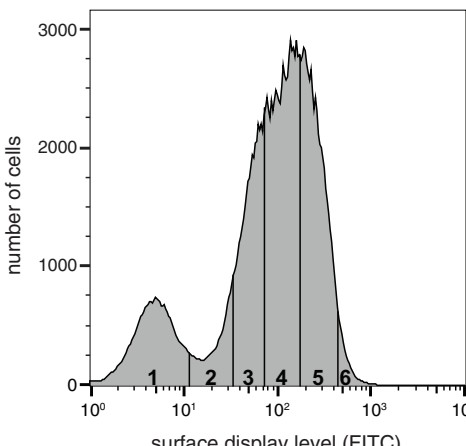

| sorting bin | mean fluorescence | fraction of total cells |
|---|---|---|
| 1 | 5 | 0.14 |
| 2 | 23 | 0.05 |
| 3 | 54 | 0.20 |
| 4 | 121 | 0.30 |
| 5 | 246 | 0.25 |
| 6 | 471 | 0.05 |

**Appendix 1—figure 1.** Distribution of surface-display levels and sorting bins for a library of peptides based on segments of LAT, SLP-76, p38α, and TCRζ.

To analyze the sequencing data, raw counts were determined for DNA corresponding to each peptide in each of the populations (5 or 6 sorted populations and one set corresponding to an input library before sorting). The frequency of each sequence in each sorted population (raw counts for that sequence divided by the total counts for that sample) was normalized to the frequency of the same sequence in the unsorted set. These normalized frequencies were then scaled by the fraction of total cells that the particular population represents (shown in the table in *Appendix 1—figure 1*). Finally, the scaled, normalized frequencies were plotted as a function of mean fluorescence in each sorting bin (*Appendix 1—figure 1*) and fit to a Gaussian curve (*Appendix 1—figure 2*). These data describe the surface-display level distributions for each individual peptide that underlie the cumulative distribution seen in *Appendix 1—figure 1*.

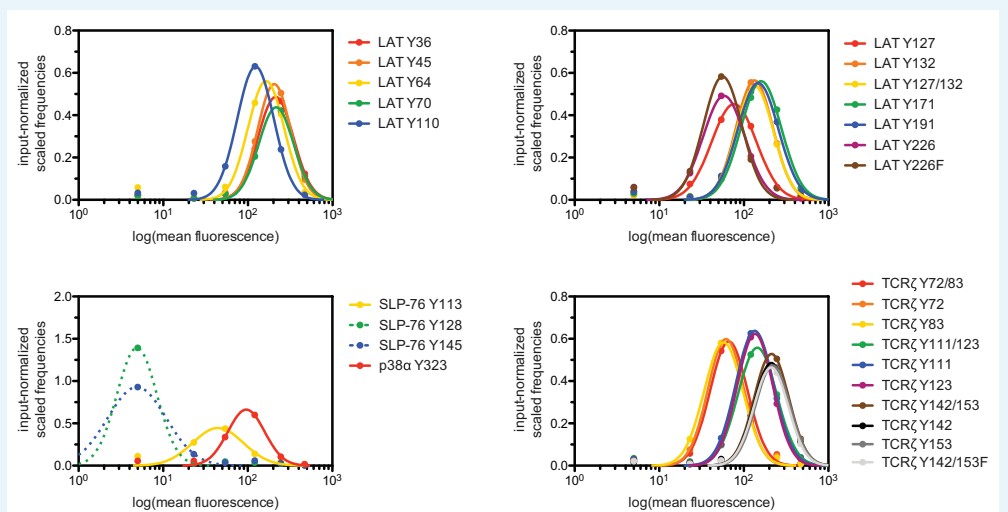

**Appendix 1—figure 2.** Surface-display distributions for individual peptides in LAT, SLP-76, p38α, and TCRζ library. Note that fits were approximated for the SLP-76 Tyr 128 and Tyr 145 peptides, as most of the cells bearing this peptide were in the lowest fluorescence bin.

From the fitted data in *Appendix 1—figure 2*, mean fluorescence values for each peptide can be extracted. These values represent the relative surface-display levels of each peptide. As shown in *Appendix 1—figure 3*, surface-display levels correlate with net charge. In particular, surface expression levels are very low for peptides corresponding to SLP-76 Tyr 128 and SLP-76 Tyr 145, which have charges of $-12$ and $-10$, respectively.

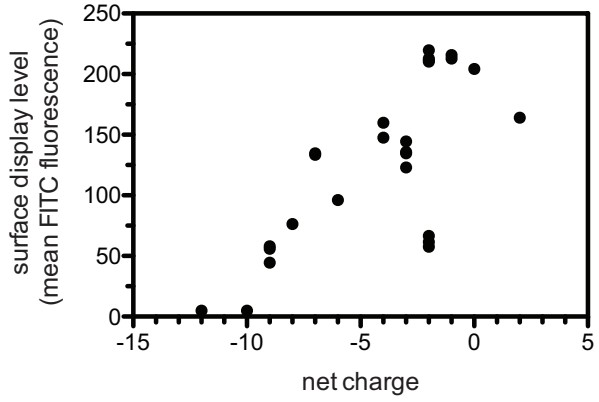

**Appendix 1—figure 3.** Correlation between surface-display level and net charge of the displayed peptide.

We carried out the data analysis described above with the scanning point mutant library for the peptide spanning Tyr 226 and normalized the mean fluorescence for each variant in this library to that of the wild-type sequence (*Appendix 1—figure 4*). Most of the point mutants in this library had expression levels within a factor of 3 of wild-type (*Appendix 1—figure 4B*). The peptides with the lowest display levels were those for which a hydrophobic residue was mutated to either aspartate or glutamate (*Appendix 1—figure 4A*). Importantly, this procedure was highly reproducible for all but the lowest-expressing peptides (*Appendix 1—figure 4C*).

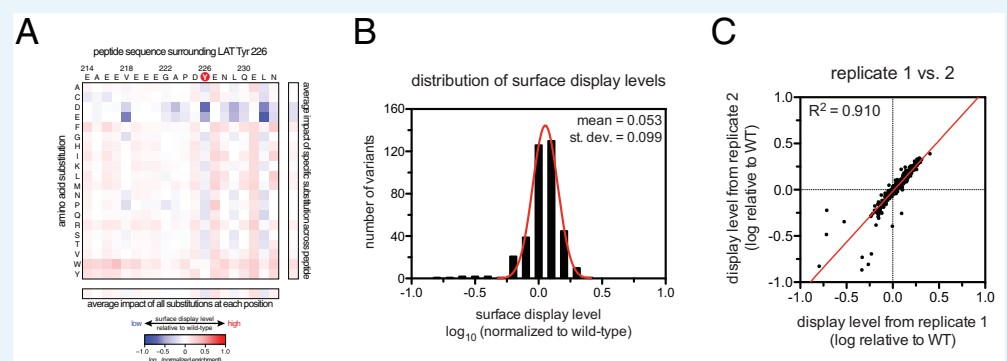

**Appendix 1—figure 4.** Analysis of surface-display levels for point mutants in the LAT Tyr 226 library. (**A**) Heat map depicting the variation in display levels for every member of the library (average values from three independent measurements). (**B**) Histogram of the distribution of display levels (average values from three independent measurements). (**C**) Correlation between calculated surface-display levels from two independent replicates.

Finally, we treated the relative surface-display level values shown in *Appendix 1—figure 4* as a normalization factor that could be used to correct for expression level biases in our phosphorylation screens. This correction procedure is summarized in *Appendix 1—figure 5*. For this scanning point mutant library, the difference in enrichment values after correction for surface-display level is small. Thus, for the analysis of all scanning point mutant libraries, we did not apply a correction for variations in surface-display.

$$\Delta E_{mut} = \log\left(\frac{freq_{mut}^{sorted}}{freq_{mut}^{input}}\right) - \log\left(\frac{freq_{WT}^{sorted}}{freq_{WT}^{input}}\right) - \log\left(\frac{display_{mut}}{display_{WT}}\right)$$

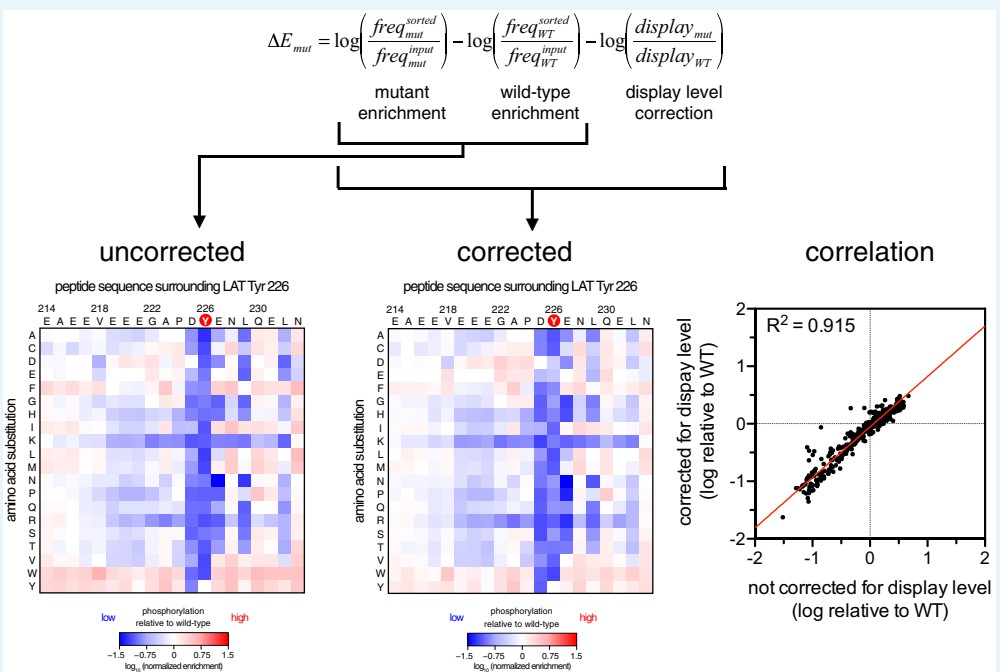

**Appendix 1—figure 5.** Correction for surface-display level in the dataset for ZAP-70 phosphorylation of the LAT Tyr 226 library. The uncorrected phosphorylation matrix is shown on the left, the phosphorylation matrix corrected for surface display level is shown in the middle, and the correlation between the uncorrected and corrected datasets is shown on the right.

