## [Decision Letter]

Thank you for submitting your article "An electrostatic selection mechanism controls sequential kinase signaling downstream of the T cell receptor" for consideration by *eLife*. Your article has been favorably evaluated by Tony Hunter as the Senior Editor and three reviewers, including Benoît Roux (Reviewer #3) and a member of our Board of Reviewing Editors.

The reviewers have discussed the reviews with one another and the Reviewing Editor has drafted this decision to help you prepare a revised submission.

Summary:

The manuscript by Shah et al. is a comprehensive analysis of the amino acid substrate specificity of two important immunologic tyrosine kinases, ZAP-70 and Lck. There have been many peptide library approaches to the primary substrate specificity of protein kinases in general and tyrosine kinases specifically, and these have generally been disappointingly inconclusive about predicting or clarifying physiologic substrate information. This manuscript though breaks new ground in generating methods and principles that seem likely to be reliable and of broad interest to the kinase field. There are several notable features. First, the manuscript convincingly confirms the role of local amino acids in influencing the specificity of ZAP-70 vs. Lck. Using a combination of bacterial display libraries and synthetic peptides, the authors make these distinctions clear and enlightening. Second, the authors uncover the structural basis for the Lck vs. ZAP-70 selectivity. This is probably the most novel and important part of the manuscript. Using computational methods, the authors predicted a basic cluster in the ZAP-70 catalytic domain, not present in Lck, which serves to repel basic residues in the vicinity of the target Tyr in substrates and is attracted to acidic residues in substrates. They elegantly confirm this using synthetic peptide substrates and cellular transfection assays. This is a triumph for molecular simulation. Third, taking these findings into account, the authors generate a model of how activation loop trans-autophosphorylation takes place. The basis of such autophosphorylation has been vexing for the field and the authors make a good case for why ZAP-70 cannot activate itself but Lck (and many other TKs) can. One of the highlights of the paper was the use of transfection experiments and chimeric tyrosine kinases that helped validate the model. Fourth, the authors use a Brownian motion simulation in a productive way to understand substrate specificity, not yet a mainstream endeavor in protein/enzyme analysis. Overall, this paper is rich with insight and novelty that should be of great interest to the readers of *eLife*, although we ask the authors to consider the points below.

Essential revisions:

The plotting of peptide kinetics vs. bacterial display preferences in Figure 5 is odd in that the kinase kinetics is presented in a linear scale and the bacterial display data on a log scale. It is not clear what the physical basis for this is. In linear free energy correlations, it is customary to use a log-log plot. For that matter, it seems surprising that the kinetic effects with kinase assays and peptides seem much smaller than the bacterial display selectivity effects. Perhaps the authors could comment on this.

The authors point out that there have not been convincing structures reported that show how activation loop tyrosine kinase phosphorylations occur in trans. The reviewers are not aware of any examples either. However, in PMID: 19060208, Chen et al. make a good case that their crystal structure of the *FGFR2* kinase domain captures trans-phosphorylation state of one kinase molecule phosphorylating the C-terminus of the other. The authors should probably cite this study and briefly describe the similarities/differences between their findings and that of the *FGFR2* structural analysis of Chen et al.

Given the strong conclusion about long-range electrostatic steering as a critical aspect of the ZAP-70-LAT interaction, it would be interesting to determine the effects of varied salt concentration on the kinase reactions. Presumably, the phosphorylation rate should increase at low salt and decrease at very high salt. This type of salt dependence of the induction phase in kinetic assays was exploited to the role of an electrostatic network in a previous study (Ozkirimli et al. Protein Sci. 2008 Nov;17(11[11]):1871-80). We recognize, however, that the authors have already included an extensive set of experiments in this manuscript. If they choose not to include additional measurements in this study, perhaps they could comment on this issue.

---

## [Author Response]

*[…] Essential revisions:*

*The plotting of peptide kinetics vs. bacterial display preferences in Figure 5 is odd in that the kinase kinetics is presented in a linear scale and the bacterial display data on a log scale. It is not clear what the physical basis for this is. In linear free energy correlations, it is customary to use a log-log plot. For that matter, it seems surprising that the kinetic effects with kinase assays and peptides seem much smaller than the bacterial display selectivity effects. Perhaps the authors could comment on this.*

The reviewers make a good point about the kinetic data being displayed on a linear scale and the enrichment values being displayed on a logarithmic scale. We have converted our graph in Figure 5 to a log-log plot. Indeed, the correlation between in vitro phosphorylation rates and enrichment values from our screen fits better to a line when both variables are log-transformed.

The difference in the magnitudes of the kinetic effects and the bacterial display selectivity effects (the slope of the line in Figure 5) is somewhat arbitrary, and it will be dependent on the screen. For example, the mutants shown in Figure 5 span a range of -1 to 1 in log_10_(enrichment), and this corresponds to a 2- to 3-fold change in phosphorylation rate. This reflects the fact that the wild-type peptide, surrounding LAT Tyr 226, is a good substrate for ZAP-70, and thus individual substitutions only have a modest, albeit measurable, effect. By contrast, for ZAP-70 phosphorylation of a poor substrate, the peptide encompassing LAT Tyr 132 (Figure 5), substitutions at Gly 131 have a log_10_(enrichment) of roughly 1, but this can correspond to a 16-fold enhancement in phosphorylation rate (Figure 5—figure supplement 4). We consider the ability to accurately detect small effects from point mutants, even in the context of an optimal substrate, to be a strength of our specificity screening platform. We have modified the text, where the results of the screens are first introduced, to state this.

*The authors point out that there have not been convincing structures reported that show how activation loop tyrosine kinase phosphorylations occur in trans. The reviewers are not aware of any examples either. However, in PMID: 19060208, Chen et al. make a good case that their crystal structure of the FGFR2 kinase domain captures trans-phosphorylation state of one kinase molecule phosphorylating the C-terminus of the other. The authors should probably cite this study and briefly describe the similarities/differences between their findings and that of the FGFR2 structural analysis of Chen et al.*

We agree that Chen et al. make a compelling case that the structure reported in their paper captures a *trans*-phosphorylation complex. We have analyzed that structure, which depicts C-terminal tail phosphorylation, along with other proposed *trans*-autophosphorylation structures that may be relevant to activation loop phosphorylation. During our analysis, we also looked at serine/threonine kinase *trans*-autophosphorylation complexes, as suggested in another reviewers’ comment. To reflect these considerations, we have modified our section entitled “A model for tyrosine kinase activation loop phosphorylation” to contain a consolidated paragraph that cites all of the appropriate references, including Chen et al., and explains why we ultimately utilized PDB code 3LVP for our modeling of the Lck and c-Src autophosphorylation complexes.

*Given the strong conclusion about long-range electrostatic steering as a critical aspect of the ZAP-70-LAT interaction, it would be interesting to determine the effects of varied salt concentration on the kinase reactions. Presumably, the phosphorylation rate should increase at low salt and decrease at very high salt. This type of salt dependence of the induction phase in kinetic assays was exploited to the role of an electrostatic network in a previous study (Ozkirimli et al. Protein Sci. 2008 Nov;17(11):1871-80). We recognize, however, that the authors have already included an extensive set of experiments in this manuscript. If they choose not to include additional measurements in this study, perhaps they could comment on this issue.*

The reviewers make a valid point about salt-dependent phosphorylation. We have measured phosphorylation of a LAT-based sequence by ZAP-70 under varying salt concentrations, and these data are now included Figure 8—figure supplement 1. As expected, increasing the ionic strength of the reaction solution dramatically reduces the rate of LAT phosphorylation. The end of the section on Brownian dynamics and long-range electrostatics has been updated to reference these new data. The citation provided by the reviewer is interesting because it points to the importance of an electrostatic network in controlling the activity of Src kinases, such as Lck, and we now cite this work in the revised manuscript.